# Learning Human-Compatible Representations for Case-Based Decision Support

**Han Liu, Yizhou Tian, Chacha Chen, Shi Feng, Yuxin Chen & Chenhao Tan**
Department of Computer Science, University of Chicago
{hanliu,tianh,chacha,shif,chenyuxin,chenhao}@uchicago.edu

## Abstract

Algorithmic case-based decision support provides examples to aid people in decision making tasks by providing contexts for a test case. Despite the promising performance of supervised learning, representations learned by supervised models may not align well with human intuitions: what models consider similar examples can be perceived as distinct by humans. As a result, they have limited effectiveness in case-based decision support. In this work, we incorporate ideas from metric learning with supervised learning to examine the importance of alignment for effective decision support. In addition to instance-level labels, we use human-provided triplet judgments to learn human-compatible decision-focused representations. Using both synthetic data and human subject experiments in multiple classification tasks, we demonstrate that such representation is better aligned with human perception than representation solely optimized for classification. Human-compatible representations identify nearest neighbors that are perceived as more similar by humans and allow humans to make more accurate predictions, leading to substantial improvements in human decision accuracies (17.8% in butterfly vs. moth classification and 13.2% in pneumonia classification).

## 1 Introduction

Despite the impressive performance of machine learning (ML) models, humans are often the final decision maker in high-stake domains due to ethical and legal concerns (Lai & Tan, 2019; Green & Chen, 2019), so ML models as decision support is preferred over full automation. In order to provide meaningful information to human decision makers, the model cannot be illiterate in the underlying problem, e.g., a model for assisting breast cancer radiologists should have a high diagnostic accuracy by itself. However, a model with high *autonomous* performance may not provide the most effective decision support, because it could solve the problem in a way that is not comprehensible or even perceptible to humans, e.g., AlphaGo's famous move 37 (Silver et al., 2016; 2017; Metz et al., 2016). Our work studies the relation between these two objectives that effective decision support must balance: achieving high autonomous performance and aligning with human intuitions.

We focus on case-based decision support for classification problems (Kolodneer, 1991; Begum et al., 2009; Liao, 2000; Lai & Tan, 2019). For each test example, in addition to showing the model's predicted label, case-based decision support shows one or more related examples retrieved from the training set. These examples can be used to justify the model's prediction, e.g., by showing similar-looking examples with the predicted label, or to help human decision makers calibrate its uncertainty, e.g., by showing similar-looking examples from other classes. Both use cases require the model to know what is similiar-looking to the human decision maker. In other words, an important consideration in aligning with human intuition is approximating human judgment of similarity.

Figure 1 illustrates the importance of such alignment on a classification problem of distinguishing butterfly from moth. A high-accuracy ResNet (He et al., 2016) produces a highly linearly-separable representation space, which leads to high classification accuracy. But the nearest neighbor cannot provide effective justification for model prediction because it looks dissimilar to the test example for humans. The similarity measured in model representation space does not align with human visual similarity. If we instead use representations from a second model trained specifically to mimic human visual similarity rather than to classify images, the nearest neighbor would provide strong

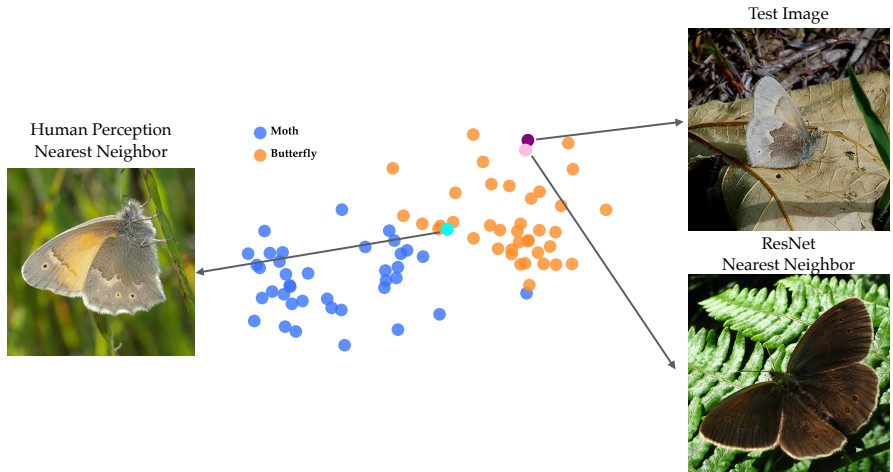

Figure 1: Nearest neighbor retrieved by the model representation might not align with human similarity judgment. The MLE representations (512-dim) are visualized using t-SNE (Van der Maaten & Hinton, 2008). The purple circle represents a specific test instance. The nearest neighbor found by MLE representations (pink circle) is not as visually similar as the instance in cyan circle found by optimizing a metric learning objective.

justification for the model prediction. However, using the second model for decision support has the risk of misleading or even deceiving the human decision maker because the "justification" is generated based on a representation space that is different from the model used to predict the label; it becomes persuasion rather than justification.

The goal of this work is to learn a *single* representation space that satisfies two properties: (i) producing easily separable representations for different classes to support accurate classification, and (ii) constituting a metric space that is aligned with human perception of similarity between examples. Simultaneously matching the best model on classification accuracy and achieving perfect approximation of human similarity might not be possible, but we hypothesize that a good trade-off between the two would benefit decision support. We propose a novel multi-task learning method that combines supervised learning and metric learning. We supplement the standard maximum likelihood objective with a triplet margin loss function from Balntas et al. (2016). Our method learns from human annotations of similarity judgments among data instances in the triplet form.

We validate our approach with both synthetic data and user study. We show that representations learned from our framework identify nearest neighbors that are perceived as more similar by the synthetic human than that based on supervised classification (henceforth *MLE representations*, see §2 for details), and are therefore more suitable to provide decision support. We further demonstrate that the advantage of human-compatible representations indeed derives from human perception rather than data augmentation.

We further conduct human subject experiments using two classification tasks: (i) butterfly vs. moth classification from ImageNet (Krizhevsky et al., 2012), and (ii) pneumonia classification based on chest X-rays (Kermany et al., 2018). Our results show that human-compatible representations provide more effective decision support than MLE representations. In particular, human-compatible representations allow laypeople to achieve an accuracy of 79.1% in pneumonia classification, 15.3% higher than MLE representations. A similar improvement has been observed on the butterfly vs. moth classification task (34.8% over MLE representations and 17.8% over random).

To summarize, our main contributions include:
- We highlight the importance of alignment in learning human-compatible representations for case-based decision support.
- We propose a multi-task learning framework that combines supervised learning and metric learning to simultaneously learn classification and human visual similarity.
- We design a novel evaluation framework for comparing representations in decision support.
- Empirical results with synthetic data and human subject experiments demonstrate the effectiveness of our approach.

## 2 CASE-BASED DECISION SUPPORT

Consider the problem of using a classification model $h : \mathcal{X} \to \mathcal{Y}$ as decision support for humans. Simply showing the predicted label from the model provides limited information. Explanations are commonly hypothesized to improve human performance by providing additional information (Doshi-Velez & Kim, 2017). We focus on information presented in the form of examples from the training data, also known as case-based decision support (Kolodneer, 1991; Begum et al., 2009; Liao, 2000; Lai & Tan, 2019). Case-based decision support can have diverse use cases and goals. Given a test example ($x$) and its predicted label ($\hat{y}$), two common use cases are:

- Presenting the nearest neighbor of $x$ along with label $\hat{y}$ as a justification of the predicted label. We refer to this scenario as *justification* (Kolodneer, 1991).
- Presenting the nearest neighbor in each class without presenting $\hat{y}$. This approach makes a best-effort attempt to provide evidence and leaves the final decision to humans, without biasing humans with the predicted label. We refer to this scenario as *neutral decision support* (Lai & Tan, 2019).

**Formulation.** Building on Kolodneer (1991), we formalize the problem of case-based decision support in the context of representation learning. The goal is to assist humans on a classification problem with groundtruth $f : \mathcal{X} \to \mathcal{Y}$. We assume access to a representation model $g$, which takes an input $x \in \mathcal{X}$ and generates an $m$-dimensional representation $g(x) \in \mathbb{R}^m$. For each test instance $x$, an example selection policy $\pi$ chooses $k$ labeled examples from the training set $D^{\text{train}}$ and shows them to the human (optionally along with the labels); the human then makes a prediction by choosing a label from $\mathcal{Y}$. As discussed in the two common use cases, we consider nearest-neighbor-based selection policies in this work. The focus of this work is thus on the effectiveness of $g$ for case-based decision support.

Given a neural classification model $h : \mathcal{X} \to \mathcal{Y}$, the representation model is the last layer before the classification head, which is a byproduct derived from $h$. We refer to this model as $e(h)$.[1] In justification, the example selection policy is $\pi = \text{NN}(x, e(h), D_{\hat{y}}^{\text{train}})$, where $\hat{y} = h(x)$, $D_{\hat{y}}^{\text{train}}$ refers to the subset of training data with label $\hat{y}$ (i.e., $\{(x, y) \in D^{\text{train}} \mid y = \hat{y}\}$), and NN finds the nearest neighbor of $x$ using representations from $e(h)$ among the subset of examples with label $\hat{y}$. In decision support, the example selection policy is $\{\text{NN}(x, e(h), D_y^{\text{train}}), \, \forall y \in \mathcal{Y}\}$.

**Misalignment with human similarity metric is detrimental.** We argue that aligning model representations with human similarity metric is crucial for case-based decision support; we refer to it as the *metric alignment problem*. To illustrate the importance of alignment, we need to reason about the goal of case-based decision support. Let us start with justification, which is a relatively easy case. To justify a predicted label, the chosen example should ideally *appear similar* to the test image. Crucially, this similarity is perceived by humans (i.e., interpretable), and the example selection policy identifies the nearest neighbor based on model representation (i.e., faithful). The gap between human representation and model representation (Fig. 1) leads to undesirable justification.

Neutral decision support, however, represents a more complicated scenario. We start by emphasizing that the goal is not simply to maximize human decision accuracy, because one may use policies that intentionally show distant examples to nudge or manipulate humans towards making a particular decision.[2] Choosing the nearest neighbors in each class is thus an attempt to present *faithful* and *neutral* evidence from the representation space so that humans can make their own decisions, hence preserving their agency. Therefore, the chosen nearest neighbors should be visually similar to the test instance by human perception, again highlighting the potential gap between model representation and human representation. Assuming that humans follow the natural strategy by picking the presented instance that's most *similar* to the test instance and answering with the corresponding label, then ideally, nearest neighbors in each class retain key information useful for classification so that they can reveal the separation learned in the model.

It is unlikely that we get high alignment by solely optimizing classification even when the model's classification accuracy is comparable to the human's. Models trained with supervised learning

---

[1] In general, we can use the representation in any layer, but in preliminary experiments, we find representation from the last layer is most effective.

[2] We will consider one such policy for the sake of evaluating the quality of representations in §3.

almost always exploit patterns in the training data that are (i) not robust to distribution shifts, and (ii) counterintuitive or even unobservable for humans (Ilyas et al., 2019; Xiao et al., 2020).

**Combining metric learning on human triplets with supervised classification.** We propose to address the metric alignment problem with additional supervision on the human similarity metric. We collect data in the form of human similarity judgment triplets (or *triplets* for short). Each triplet is an ordered tuple: $(x^r, x^+, x^-)$, which indicates $x^+$ is judged by humans as being closer to the reference $x^r$ than $x^-$ (Balntas et al., 2016). Given a triplet dataset $T$ and labeled classification dataset $D$, we learn a model $\theta$ using triplet margin loss (Balntas et al., 2016) in conjunction with cross-entropy loss, controlled by a hyperparameter $\lambda$:

$$\lambda \underbrace{\left[ - \sum_{(x,y) \sim D} \log\left(p_\theta(y|x)\right) \right]}_{\text{Cross-entropy loss}} + (1-\lambda) \underbrace{\left[ \sum_{(x^r, x^+, x^-) \sim T} \max\left(d_\theta(x^r, x^+) - d_\theta(x^r, x^-) + 1, 0\right) \right]}_{\text{Triplet margin loss}} \quad (1)$$

where $d_\theta(\cdot, \cdot)$ is the similarity metric based on model representations; we use Euclidean distance. In this work, we initialize $\theta$ with a pretrained ResNet (He et al., 2016). When $\lambda = 1$ and the triplet margin loss is turned off, the model reduces to a finetuned ResNet. When $\lambda = 0$ and the cross-entropy loss is turned off, the model reduces to the triplet based-learning model of Balntas et al. (2016); we call it `TMLModel` and will use it to simulate humans in some synthetic experiments in the appendix. Our work is concerned with the representations learned by these models. Our approach uses the representations learned with $\lambda = 0.5$ (henceforth *human-compatible representations* and `HC` for short). We refer to the representations fine-tuning ResNet with the cross-entropy loss as *MLE representations* (`MLE` for short) and the representations from `TMLModel` as `TML`.

## 3 EXPERIMENTAL SETUP

In this section, we provide the specific model instantiation and detailed experiment setup.

**Models.** All models and baselines use ResNet-18 (He et al., 2016) pretrained on ImageNet as the backbone image encoder. Following Chen et al. (2020), we take the output of the average pooling layer and feed it into an MLP projection head with desired embedding dimension. We use the output of the projection head as our final embeddings (i.e., representations), where we add task-specific head and loss for training and evaluation. We use Euclidean distance as the similarity metric for both loss calculation and distance measurement during example selection in decision support.

Our first baseline uses representations from ResNet finetuned with classification labels using cross-entropy loss (i.e., `MLE`). ResNet typically achieves high classification accuracy but does not necessarily produce human-aligned representations. Our second baseline uses representations from the same pretrained model finetuned with human triplets using triplet margin loss (Balntas et al., 2016) (i.e., `TML`). We expect `TML` to produce more aligned representations but achieve lower classification accuracy than `MLE` and may provide limited effectiveness in decision support.

Our representations, `HC`, are learned by combining the two loss terms following Equation 1. The hyperparameter $\lambda$ controls the trade-off between metric alignment and classification accuracy: with higher $\lambda$ we expect `HC` to be more similar to `MLE`, while lower $\lambda$ steers `HC` towards `TML`. Empirically tuning $\lambda$ confirms this hypothesis. For the main paper, we present results with $\lambda = 0.5$. More details about model specification and hyperparameter tuning can be found in the appendix.

**Filtering classification-inconsistent triplets.** Human triplets may not always align with classification: triplet annotators may choose the candidate from the incorrect class over the one from the correct class. We refer to these data points as *classification-inconsistent triplets*. We consider a variant of human-compatible representations where we isolate human intuition that's compatible with classification and remove these classification-inconsistent triplets from the training set; we refer to this condition as `HC-filtered`. Filtering is yet another way to strike a balance between human intuition and classification. We leave further details on filtering in the appendix.

**Evaluation metrics.** Our method is designed to align representations with human similarity metrics and at the same time retain the representations' predictive power for classification. We can evaluate these representations with classification and triplet accuracy using existing data, but our main evaluation is designed to simulate case-based decision support scenarios.

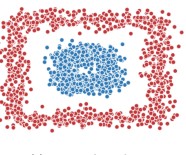 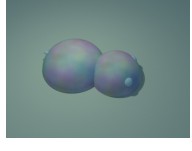 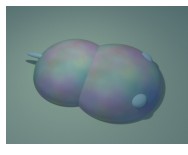 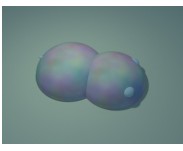 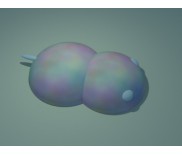

(a) Decision boundary      (b) Vespula 1      (c) Vespula 2      (d) Weevil 1      (e) Weevil 2

Figure 2: VW dataset. (a) shows the dataset where labels are determined (non-linearly) by two features: the head and the body size of the fictional insects. (b)-(d) show samples of the two classes; the Weevil has a mid-sized body and mid-sized head, while the Vespula does not. Tail length and texture are two non-informative features.

- **Head-to-head comparisons** ("**H2H**"). To evaluate justification, we set up head-to-head comparisons between two representations ($R_1$ vs. $R_2$) and ask: given a test instance and two justifications retrieved by $R_1$ and $R_2$, which justification do humans consider as closer to the test instance? We report the fraction of rounds that $R_1$ is preferable. In addition to the typical justification for the predicted label, we also examine that for classes other than the predicted class, as those examples will be used in decision support for users to examine the plausibility of each class. We refer to the nearest example in the *predicted* class as *NI*, and the nearest example in the other class as *NO*.
- **Neutral decision support**. Following §2, we retrieve the nearest neighbors from each class. We use the accuracy of humans as the measure of effective decision support.
- **Persuasive decision support**. We retrieve the nearest example with the predicted label and the furthest example from the other class. If the representation is aligned with human similarity metric, this approach encourages people to follow the predicted label, which likely leads to over-reliance and may be unethical in practice. Here, we use this scenario as a surrogate to evaluate the quality of the learned representations.

Note that we do not show model predictions so that humans focus on the similarity between examples.

## 4 SYNTHETIC EXPERIMENT

To understand the strengths and limitations of our method, we first experiment with synthetic datasets. Using simulated human similarity metrics, we control and vary the level of disagreement between the classification groundtruth and the synthetic human's knowledge.

### 4.1 SYNTHETIC DATASET AND SIMULATED HUMAN SIMILARITY METRICS

We use the synthetic dataset "Vespula vs Weevil" (VW) from Chen et al. (2018b). It is a binary image classification dataset of two fictional species of insects. Each example contains four features, two of them—head and body size—are predictive of the label, and the other two—tail length and texture—are completely non-predictive. We generate 2000 images and randomly split the dataset into training, validation, and testing sets in a 60%:20%:20% ratio. The labels are determined by various synthetic decision boundaries, such as the one shown in Fig. 2a.

To generate triplets data, we define simulated human similarity metrics as a weighted Euclidean distance over the visual features: for any instance $a$ and $b$, $d(a, b) = \sqrt{\sum_i w_i (a_i - b_i)^2}$, where $i$ refers to the $i$-th feature. By changing the weight of each feature, we can control the level of disagreement between a synthetic human and the groundtruth. All procedures that involve humans (i.e., triplet data collection and evaluation) are done by the synthetic human in this section.

To quantify the disagreement, we use 1-NN classification accuracy following the synthetic human similarity metric; we refer to it as the *task alignment score*. Note that this is different from our main alignment problem, which is about the representations. The task alignment score ranges from 50% (setting the informative features' weights to 0 and distractor weights to 1) to 100%. See the appendix for more details on how we generate these weights. In each setting, we generate 40,000 triplets.

### 4.2 RESULTS

We compare `HC`, `MLE`, `TML` on classification accuracy, triplet accuracy, and decision support performance for the synthetic human. We train all three representations with a large dimension of 512 and

Table 1: Experiment results on VW with H2H comparison and decision support evaluations.

| Task alignment | 50% | 80% | 83% | 92% | 92.5% | 100% |
|---|---|---|---|---|---|---|
| Weights | [0,0,1,1] | [1,0,1,1] | [0,1,1,1] | [1,256,256,256] | [256,1,256,256] | [1,1,1,1] |
| **NI-H2H** | | | | | | |
| HC vs. MLE | 0.917 | 0.914 | 0.903 | 0.880 | 0.872 | 0.808 |
| **NO-H2H** | | | | | | |
| HC vs. MLE | 0.916 | 0.968 | 0.946 | 0.958 | 0.962 | 0.970 |
| **Neutral decision support** | | | | | | |
| MLE | 0.753 | 0.899 | 0.896 | 0.897 | 0.901 | 0.929 |
| TML | 0.568 | 0.775 | 0.807 | 0.868 | 0.877 | **1.000** |
| HC | **0.759** | **0.901** | **0.928** | **0.949** | **0.955** | **1.000** |
| **Persuasive decision support** | | | | | | |
| MLE | 0.704 | 0.900 | 0.903 | 0.903 | 0.901 | 0.919 |
| TML | 0.906 | 0.881 | 0.863 | 0.876 | 0.877 | **1.000** |
| HC | **1.000** | **1.000** | **1.000** | **1.000** | **1.000** | **1.000** |

a small dimension of 50 and observe that the 512-dimension representation is preferable based on most metrics. We also train HC on filtered vs. unfiltered triplets as well as with different values $\lambda$. For our main results, we report the performance with $\lambda = 0.5$ and filtered triplets for the decision boundary in Fig. 2a. We will discuss the effect of filtering later in this section. $\lambda$'s role is relatively limited and we will discuss its effect and other decision boundaries in the appendix.

In synthetic experiments, HC achieves the same perfect classification accuracy as MLE (100%), and a triplet accuracy of 96.8%, which is comparable to TML (97.3%). This shows that HC indeed learns both the classification task and human similarity prediction task. We next present the evaluation of case-based decision support with the synthetic human, which is the key goal of this work.

**HC significantly outperforms MLE in H2H.** If there is no difference between HC and MLE, the synthetic human should prefer HC about 50% of times. However, as shown in Table 1, our synthetic human prefer HC over MLE by a large margin (about 90% of times) as justifications for both nearest in-class examples and nearest out-of-class examples, indicating the NIs and NOs selected based on the HC representations are more aligned with the synthetic human than MLE. For NI H2H, the preference towards HC declines as the task alignment improves, because if alignment between human similarity and classification increases, MLE can capture human similarity as a byproduct of classification.

**HC provides the best decision support.** Table 1 shows that HC achieves the highest neutral and persuasive decision support accuracies in all task alignments. In neutral decision support, MLE consistently outperforms TML, highlighting that representation solely learned for metric learning is ineffective for decision support. For all models, the decision support performance improves as the task alignment increases, suggesting that decision support is easier when human similarity judgment is aligned with the classification task. MLE and TML are more comparable in persuasive decision support, while HC consistently achieves 100%. The fact that MLE shows comparable performance between neutral and persuasive decision support further confirms that MLE does not capture human similarity for examples from different classes.

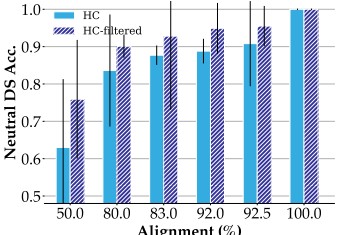

Figure 3: Neutral decision support with HC and HC-filtered. HC-filtered leads to improved performance.

**Filtering triplets leads to better decision support**. Fig. 3 shows that filtering class-inconsistent triplets improves HC's decision support performance across all alignments. Further details in the appendix show that filtering slightly hurts H2H performance. This suggests that in terms of decision support, the benefit of filtering out human noise may outweigh the loss of some similarity judgment.

**The importance of human perception.** One may question whether filtering class-inconsistent triplets essentially provides additional label supervision in the form of triplets. We show this is not the

Table 2: Experiment results on VW using synthetic human with 92% alignment. Comparing MLE representations and `HC-filtered` with `HC` trained on label-derived triplets and `HC` trained on same-class triplets. 40,000 new triplets were generated for each condition.

| Evaluations | MLE | HC label-derived triplets | HC same-class triplets | HC-filtered |
|---|---|---|---|---|
| NI-H2H with MLE | N/A | 0.509 | 0.890 | 0.889 |
| NO-H2H with MLE | N/A | 0.607 | 0.970 | 0.958 |
| Neutral DS | 0.897 | 0.723 | 0.960 | 0.949 |
| Persuasive DS | 0.903 | 0.803 | 0.998 | 1.000 |

case by experimenting with `HC` trained on *label-derived triplets*. Assuming that an instance is more similar to another instance with the same label than one with a different label, we derive label-derived triplets directly from groundtruth labels ($x^+$ from the same class as $x^r$ and $x^-$ from the other class), containing no human perception information. Table 2 shows decision support results for this setting: `HC` label-derived triplets show worse performance than `HC-filtered`. In fact, `HC` label-derived triplets show even worse neutral and persuasive decision support than `MLE`, which may be due to label-derived triplets causing overfitting. This suggests that triplets without human perception do not lead to human-compatible representations.

We also experiment with `HC` trained on same-class triplets, human-triplets but only those where the non-reference cases ($x^+, x^-$) are from the same class; that is, the triplets cannot provide any label supervision. We observe from Table 2 that `HC` trained on these triplets show similar results to `HC-filtered` across all decision support evaluations. This suggests that human perception is the main factor in driving human-compatible representations' high decision support performance.

## 5 HUMAN SUBJECT EXPERIMENTS

We conduct human subject experiments on two image classification datasets: a natural image dataset, Butterflies v.s. Moths (BM) and a medical image dataset of chest X-rays (CXR). For BM, we followed Singla et al. (2014) and acquired 200 images from ImageNet (Krizhevsky et al., 2012). BM is a binary classification problem and each class contains two species. CXR is a balanced binary classification subset taken from Kermany et al. (2018) with 3,166 chest X-ray images that are labeled with either normal or pneumonia. We randomly split the datasets following 60%:20%:20% ratio. The classification accuracy with our base supervised learning models are 97.5% for BM and 97.3% for CXR. We only present results with human subjects in the main paper, but results from simulation experiments with `TML` as a synthetic agent, such as filtering triplets providing better results, are qualitatively consistent. See §D and §E in the appendix for more details.

### 5.1 TRIPLET ANNOTATION

We recruit crowdworkers on Prolific to acquire visual similarity triplets. In each question, we show a reference image on top and two candidate images below, and ask a 2-Alternative-Forced-Choice (2AFC) question: which candidate image looks more similar to the reference image? A screenshot of the interface can be found in the appendix. To generate triplets for annotation, we first sample the reference image from either the training, the validation, or the test set. Then for each reference image, we sample two candidates from the training set. We sample the candidates only from the training set because in decision support, the selected examples should always come from the training set, and thus we only need to validate and test triplet accuracies with candidates from the training set.

For BM we recruit 80 crowdworkers, each completing 50 questions, giving us 4000 triplets. For CXR we recruit 100 crowdworkers, each answering 20 questions, yielding 2000 triplets. Our pilot study suggests that visual similarity judgment on chest X-rays is a more mentally demanding task, so we decrease the number of questions for each CXR survey.

### 5.2 RESULTS ON BUTTERFLIES V.S. MOTHS

We recruit crowdworkers on Prolific to evaluate representations produced by our models by doing decision support tasks. We acquire examples with different example selection policies from `HC`

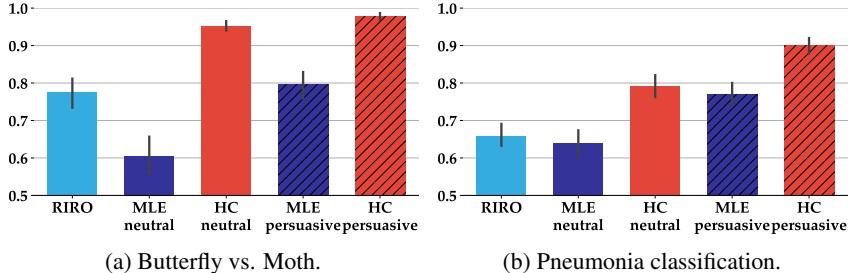

(a) Butterfly vs. Moth.  (b) Pneumonia classification.

Figure 4: Decision support accuracy with human subject studies. Error bars show 95% confidence intervals. `HC` dominates `MLE` in both neutral and persuasive decision support.

and `MLE`. We choose the dimension and training triplets of the representation based on the models' classification accuracy, triplet accuracy, and decision support simulation results based on synthetic agents. See more details in the appendix. We do not include `TML` in human studies, because in practice, `TML` models cannot make predictions on class labels, therefore are unable to distinguish and select in-class and out-of-class examples and thus cannot be used for decision support.

**H2H comparison results show `HC` NI examples are slightly but significantly preferred over `MLE` NI examples according to human visual similarity.** We recruit 30 Prolific workers to make H2H comparisons between `HC` NI examples and `MLE` NI examples over the entire test set. The mean preference for `HC` over `MLE` is 0.5316 with a 95% confidence interval of $\pm 0.0302$ ($p = 0.0413$ with one-sample t-test). This means the `HC` NI examples are closer to the test images than `MLE` NI examples with statistical significance according to human visual similarity.

**Decision support results show `HC` is significantly better than `MLE` both in neutral and persuasive decision support.** Combining two example selection policies with two representations, we have four conditions: `HC` neutral, `HC` persuasive, `MLE` neutral, `MLE` persuasive. We also add a baseline condition with random supporting examples, which we call random in-class random out-of-class (RIRO). We recruit 30 Prolific workers for each condition and ask them to go through the images in the test set with supporting examples from each class in the training set. Both the order of the test images and the order of the supporting images within each test question are randomly shuffled.

Figure 4a shows the human classification accuracies with different decision support scenarios and different representations. In neutral decision support, we observe that `HC` achieves much higher accuracy than `MLE` (95.3% vs. 60.5%, $p = 4\mathrm{e}{-19}$ with two-sample t-test). In fact, even RIRO provides better decision support than MLE representations, suggesting that the supporting images based on `MLE` are confusing and hurt human decision making (77.5% vs. 60.5%, $p = 3\mathrm{e}{-6}$). As expected, the accuracies are generally higher in persuasive decision support. `HC` enables an accuracy of 97.8%, which is much better than `MLE` at 79.5% ($p = 2\mathrm{e}{-13}$). `HC` in neutral decision support already outperforms `MLE` in persuasive decision support. These findings confirm our results with VW synthetic experiments that human-compatible representations provide much better decision support than MLE representations.

### 5.3 RESULTS ON CHEST X-RAYS

We use the same experimental setup as BM to evaluate `HC` and `MLE` representations in CXR.

**H2H comparison results show `HC` NI examples are slightly preferred over `MLE` NI examples but the difference is not statistically significant.** We recruit 50 Prolific workers to each make 20 H2H comparisons between `HC` NI examples and `MLE` NI examples. The mean preference for `HC` over `MLE` is 0.516 with a 95% confidence interval of $\pm 0.0725$ ($p = 0.379$ with one-sample t-test). H2H comparison in CXR is especially challenging as laypeople need to differentiate between two chest X-rays in the same class, hence the slightly worse performance in H2H compared to BM.

**Similar to BM, `HC` outperforms `MLE` in both neutral and persuasive decision support in CXR.** As expected, Fig. 4b shows that pneumonia classification is a much harder task than butterfly vs. moth classification, indicated by the lower accuracies across all conditions. In neutral decision support, `HC` enables much better accuracy than `MLE` (79.1% vs. 63.8%, $p = 2\mathrm{e}{-8}$ with two-sample t-test). In fact, similar to the BM setting, `MLE` provides similar performance with RIRO (63.8% vs. 65.9%,

$p = 0.390$), suggesting that MLE representations are no different from random representations for selecting nearest neighbors within a class. To contextualize our results, we would like to highlight that our crowdworkers are laypeople and have no medical training. It is thus impressive that human-compatible representations enable an accuracy of almost 80% in neutral decision support, which demonstrates the potential of human-compatible representations.

In persuasive decision support, HC provides the highest decision support accuracy at 90.0%, also much higher than MLE at 77.0% ($p = 2e-10$). Again, while we do not recommend persuasive decision support as a policy for decision support in practice, these results show that our human-compatible representations are indeed more compatible with humans than MLE representations.

## 6 RELATED WORK

**Ordinal embedding.** The ordinal embedding problem (Ghosh et al., 2019; Van Der Maaten & Weinberger, 2012; Kleindessner & von Luxburg, 2017; Kleindessner & Luxburg, 2014; Terada & Luxburg, 2014; Park et al., 2015) seeks to find low-dimensional representations that respect ordinal feedback. Currently, there exist several techniques for learning ordinal embeddings. Generalized Non-metric Multidimensional Scaling (Agarwal et al., 2007) takes a max-margin approach by minimizing hinge loss. Stochastic Triplet Embedding (Van Der Maaten & Weinberger, 2012) assumes the Bradley-Terry-Luce noise model (Bradley & Terry, 1952; Luce, 1959) and minimizes logistic loss. The Crowd Kernel (Tamuz et al., 2011) and t-STE (Van Der Maaten & Weinberger, 2012) propose alternative non-convex loss measures based on probabilistic generative models. These results are primarily empirical and focus on minimizing prediction error on unobserved triplets. In principle, one can plugin these approaches in our framework as alternatives to the triplet margin loss in Eq. 1.

**AI explanations and AI-assisted decision making.** Various explanation methods have been developed to explain black-box AI models (Guidotti et al., 2018), such as feature importance (Ribeiro et al., 2016; Shrikumar et al., 2017), saliency map (Zhou et al., 2016; Selvaraju et al., 2017), and decision rules (Ribeiro et al., 2018). Example-based explanations are also a type of common explanation methods that use examples to explain AI models. Nearest-neighbor examples can explain a model's local decision (Wang & Yin, 2021; Nguyen et al., 2021; Taesiri et al., 2022; Lai & Tan, 2019). To the best of our knowledge, there has been no prior work that examines the role of representations in choosing the nearest neighbors in the context of AI explanations. Meanwhile, global example-based explanations such as prototypes can explain a model's global behavior or a model's understanding of the data distribution (Kim et al., 2016; Chen et al., 2018a; Cai et al., 2019a; Lai et al., 2020). Explaining a model's global behavior is also closely related to machine teaching (Zhu et al., 2018).

Many of these explanation methods have been used in AI-assisted decision making to explain AI predictions or inform users about the AI model or training data (Lai et al., 2021). Among them, example-based explanations have shown be useful in many high-stake domains where full AI automation is often not desired, such as recidivism prediction (Hayashi & Wakabayashi, 2017) and medical diagnosis (Cai et al., 2019b; Rajpurkar et al., 2020; Tschandl et al., 2020). While many of the current literature in AI-assisted decision making focus on generating explanations of AI without considering human feedback, our decision support methods offer assistance by learning from human perceptions and provide examples from human-compatible representations.

## 7 CONCLUSION

Our work formulates the novel problem of learning human-compatible representations for case-based decision support. As we identify in this paper, the key to providing effective case-based support with a model is the alignment between the model and the human in terms of similarity metrics: two examples that appear similar to the model should also appear similar to the human. But models trained to perform classification do not automatically produce representations that satisfy this property. To address this issue, we propose a multi-task learning method to combine two sources of supervision: labeled examples for classification and triplets of human similarity judgments. With synthetic experiments and user studies, we validate that human-compatible representations (i) consistently get the best of both worlds in classification accuracy and triplet accuracy, (ii) select visually more similar examples in head-to-head comparisons, (iii) and provide better decision support.

## ACKNOWLEDGMENTS

We thank the anonymous reviewers for their insightful comments. We also thank members of the Chicago Human+AI lab for their thoughtful feedback. This work was supported in part by a CDAC discovery grant at the University of Chicago and an NSF grant, IIS-2040989.

## ETHICS STATEMENT

Although coming from a genuine goal to improve human-AI collaboration by aligning AI models with human intuition, our work may have potential negative impacts for the society. We discuss these negative impacts from two perspectives: the multi-task learning framework and the decision support policies.

### MULTI-TASK LEARNING FRAMEWORK

Our human-compatible representations models are trained with two sources of data. The first source of data is classification annotations where groundtruth maybe be derived from scientific evidence or crowdsourcing with objective rules or guidelines. The second source of data is human judgment annotations where groundtruth is probably always acquired from crowdworkers with subjective perceptions. When our data is determined with subjective perceptions, the model that learns from it may inevitably develop bias based on the sampled population. If not carefully designed, the human judgment dataset may contain bias against certain minority group depending on the domain and the task of the dataset. For example, similarity judgment based on chest X-rays of patients in one gender group or racial group may affect the generalizability of the representations learned from it, and may lead to fairness problems in downstream tasks. It is important for researchers to audit the data collection process and make efforts to avoid such potential problems.

### DECISION SUPPORT POLICIES

Among a wide variety of example selection policies, our policies to choose the decision support examples are only attempts at leveraging AI model representations to increase human performance. We believe that they are reasonable strategies for evaluating representations learned by a model, but future work is required to establish their use in practice.

The neutral decision support policy aims to select the nearest examples in each class, therefore limiting the decision problem to a small region around the test example. We hope this policy allow human users to zoom in the local neighborhood and scrutinize the difference between the relatively close examples. In other words, neutral decision support help human users develop a local decision boundary with the smallest possible margin. This could be useful for confusing test cases that usually require careful examinations. However, the neutral decision support policy adopts an intervention to present a small region in the dataset and may downplay the importance of global distribution in human users' decision making process.

The persuasive decision support policy aims to select the nearest in-class examples but the furthest out-of-class examples. It aims to maximize the visual difference between examples in opposite class, thus require less effort for human users to adopt case-based reasoning for classification. It also helps human users to develop a local decision boundary with the largest possible margin. However, when model prediction is incorrect, the policy end up selecting the furthest in-class examples with the nearest out-of-class examples, completely contrary to what it is design to do, may lead to even over-reliance or even adversarial supports.

In general, decision support policies aim to choose a number of supporting examples without considering some global properties such as representativeness and diversity. While aiming to reduce humans' effort required in task by encouraging them to make decision in a local region, the decision support examples do not serve as a representative view of the whole dataset, and may bias human users to have a distorted impression of the data distribution. It remains an open question that how to ameliorate these negative influence when designing decision support interactions with case-based reasoning.

## REPRODUCIBILITY STATEMENT

Implementation details and computing resources are documented in §B in the appendix. Hyperparameters and model configuration are reported in both the main paper and the appendix along with each experiments. Our code and data are available at `https://github.com/ChicagoHAI/learning-human-compatible-representations`.

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

# A    LIMITATIONS

We discuss some of the limitations in our work.

**Limitations of decision support policies.**    Our decision support policies are simple first steps towards a more general example selection policy for decision support. There are certain limitations of our selection policies. For example in this work, we only look at selecting two examples from the two classes in binary image classifcation tasks. We encourage future work to explore more selection methods towards effective decision-support.

In addition to the ethical concerns discussed in the main paper and the ethics statement, our neutral decision support and persuasive decision support policies have different limitations and use cases. Neutral decision support selects the nearest example from each class. Therefore when a test example lies too close to the decision boundary, the test example, in-class example, and out-of-class example may appear too similar to be distinguished by humans. This is where we may need to select examples further away with different features so that users are more likely to spot the distinction. Persuasive decision support selects the most similar example in the predicted class and the least similar example in the other class, the latter of which has a risk of being an outlier. This may invite biases about the data distribution of the other class and degrade effectiveness of decision support.

**Limitations of experimenting with crowdworkers.**    There are several limitations of experimenting with crowdworkers. First, crowdworkers may not invest as much time as domain experts in the tasks. Therefore, collected triplets may come from superficial or the salient features among the images. Second, crowdworkers or in general lay people have limited domain knowledge such as basic anatomy of body parts when working with medical image. Therefore it is less likely for them to notice the most important feature in the images. In our CXR task, we mitigate this limitation by providing an instruction and quiz section before our main study that provides basic information about how to examine chest X-rays. However, in other tasks, we may need to provide more detailed instructions and quizzes to help crowdworkers understand the task and in this way polish collected triplets.

As the expertise level of the end users increases, HC should be able to learn a high-quality representation. The effectiveness of our decision support methods may vary due to experts strong domain knowledge, but we would still expect our human-compatible representation to provide more effective decision support than MLE representations.

Our ultimate goal is to apply our method to domain experts. We start with crowdworkers and the positive results are encouraging. We hope these results could be used to convince and invite more domain experts to get involved and work towards an applicable system together in the future.

**Limitations of design choices in the algorithm.**    A number of decision choices were made in the algorithm. For example, we use Euclidean distance as the distance metric to be learned for the representation space. Experimenting with different kinds of metrics (e.g., in the psychology literature) and exploring the effectiveness of their respective representations in decision support would be an interesting future direction.

We used ResNet as the backbone network for feature extraction of images due to its competitiveness and popularity. Although model architecture is not the main concern of this paper, one could also plug in other common backbones such as DenseNet (Huang et al., 2017) and ViT (Dosovitskiy et al., 2021) into our representation learning algorithm. We leave the exploration of additional architecture and the effectiveness of their learned representation on decision support to future work.

# B    IMPLEMENTATION DETAIL

The architecture of our model is presented in Fig. 5. We first encode image inputs using a Convolutional Neural Network (CNN), and then project the output into an high-dimension representation space with a projection head made of multi-layer perceptron (MLP). In our experiments we use one non-linear layer to project the output of the CNN into our representation space. For classifcation task we add an MLP classifier head. We also use one non-linear layer with softmax activation. For triplet prediction, we re-index the representations with the current triplet batch and calculate prediction or loss. We use the PyTorch framework (Paszke et al., 2019) and the PyTorch Lightning framework

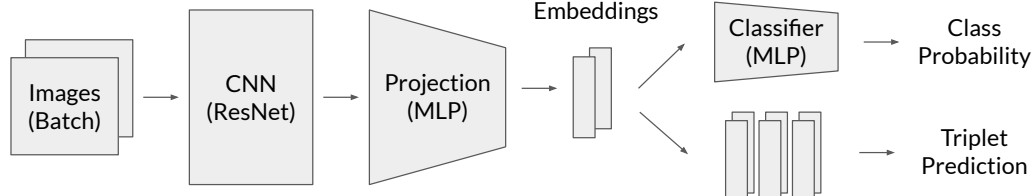

Figure 5: Architecture of the human-compatible representations model.

Table 3: Classification and triplet accuracy of human-compatible representations with different $\lambda$. `TMLModel` has no classfication head and no classification accuray.

| Model | Classification accuracy | Triplet accuracy |
|---|---|---|
| MLE | $0.998 \pm 0.003$ | $0.673 \pm 0.014$ |
| HC $\lambda = 0.8$ | $0.998 \pm 0.032$ | $0.970 \pm 0.024$ |
| HC $\lambda = 0.5$ | $0.995 \pm 0.000$ | $0.972 \pm 0.004$ |
| HC $\lambda = 0.2$ | $0.996 \pm 0.016$ | $0.973 \pm 0.039$ |
| TML | N/A | $0.973 \pm 0.016$ |

(Falcon et al., 2019) for implementation. Hyperparameters will be reported in §C for models in the synthetic experiments and in §D and §E for models in the human experiments.

### B.1 COMPUTATION RESOURCES

We use a computing cluster at our institution. We train our models on nodes with different GPUs including Nvidia GeForce RTX2080Ti, Nvidia GeForce RTX3090, Nvidia Quadro RTX 8000, and Nvidia A40. All models are trained on one allocated node with one GPU access.

## C SYNTHETIC EXPERIMENT RESULTS

### C.1 HYPERPARAMETERS

For our `MLE` backbone we use We use different controlling strength between classification and human judgment prediction, including $\lambda$s at 0.2, 0.5, and 0.8, and discuss the effect of $\lambda$ in the next section. In contrast to the experiments on BM, we observe that human-compatible representations with 512-dimension embedding shows overall better performance than human-compatible representations with 50-dimension embedding and show results for the latter in the next section. We use the Adam optimizer (Kingma & Ba, 2014) with learning rate $1e - 4$. We use a training batch size of $40$ for triplet prediction, and $30$ for classification.

### C.2 ADDITIONAL RESULTS

**Classification and triplet accuracy.** Table 3 shows how tuning $\lambda$ affects human-compatible representations's classification and triplet accuracy. Higher $\lambda$ drives human-compatible representations to behave more simlar to MLE representations while lower human-compatible representations is more similar to `TMLModel`.

**Experiment results on VW with confidence intervals.** Table 4 presents results on VW with human-compatible representations $\lambda = 0.5$. This is is simply Table 1 in the main paper with 0.95 confidence intervals.

**Results for different $\lambda$.** In Table 5 and Table 6 we show experiment results with human-compatible representations using $\lambda = 0.2$ and $\lambda = 0.8$. We do not observe a clear trend between $\lambda$ and evaluation metric performances. In the main paper we present human-compatible representations with $\lambda = 0.5$ as it shows best overall performance.

Table 4: Experiment results on VW. Models use 512-dimension embeddings; HC uses $\lambda = 0.5$ and filtered triplets. This is the same table as Table 1 and adds confidence intervals.

| Alignments | 50% | 80% | 83% | 92% | 92.5% | 100% |
|---|---|---|---|---|---|---|
| Weights | [0,0,1,1] | [1,0,1,1] | [0,1,1,1] | [1,256,256,256] | [256,1,256,256] | [1,1,1,1] |
| **NI-H2H** | | | | | | |
| HC vs. MLE | $0.917 \pm 0.064$ | $0.914 \pm 0.007$ | $0.903 \pm 0.016$ | $0.880 \pm 0.022$ | $0.872 \pm 0.020$ | $0.808 \pm 0.017$ |
| **NO-H2H** | | | | | | |
| HC vs. MLE | $0.916 \pm 0.093$ | $0.968 \pm 0.011$ | $0.946 \pm 0.009$ | $0.958 \pm 0.031$ | $0.962 \pm 0.008$ | $0.970 \pm 0.008$ |
| **Neutral decision support** | | | | | | |
| MLE | $0.753 \pm 0.056$ | $0.899 \pm 0.025$ | $0.896 \pm 0.044$ | $0.897 \pm 0.045$ | $0.901 \pm 0.025$ | $0.929 \pm 0.028$ |
| TML | $0.568 \pm 0.049$ | $0.775 \pm 0.084$ | $0.807 \pm 0.038$ | $0.868 \pm 0.012$ | $0.877 \pm 0.025$ | $\mathbf{1.000} \pm 0.000$ |
| HC | $\mathbf{0.759} \pm 0.080$ | $\mathbf{0.901} \pm 0.016$ | $\mathbf{0.928} \pm 0.099$ | $\mathbf{0.949} \pm 0.034$ | $\mathbf{0.955} \pm 0.027$ | $\mathbf{1.000} \pm 0.00$ |
| **Persuasive decision support** | | | | | | |
| MLE | $0.704 \pm 0.028$ | $0.900 \pm 0.017$ | $0.903 \pm 0.017$ | $0.903 \pm 0.017$ | $0.901 \pm 0.017$ | $0.919 \pm 0.016$ |
| TML | $0.906 \pm 0.011$ | $0.881 \pm 0.043$ | $0.863 \pm 0.044$ | $0.876 \pm 0.027$ | $0.877 \pm 0.076$ | $\mathbf{1.000} \pm 0.000$ |
| HC | $\mathbf{1.000} \pm 0.000$ | $\mathbf{1.000} \pm 0.000$ | $\mathbf{1.000} \pm 0.000$ | $\mathbf{1.000} \pm 0.000$ | $\mathbf{1.000} \pm 0.000$ | $\mathbf{1.000} \pm 0.000$ |

Table 5: Experiment results on VW. Models using 512-dimension embeddings; HC uses $\lambda = 0.2$ and filtered triplets.

| Alignments | 50% | 80% | 83% | 92% | 92.5% | 100% |
|---|---|---|---|---|---|---|
| Weights | [0,0,1,1] | [1,0,1,1] | [0,1,1,1] | [1,256,256,256] | [256,1,256,256] | [1,1,1,1] |
| **NI-H2H** | | | | | | |
| HC vs. MLE | $0.920 \pm 0.005$ | $0.890 \pm 0.032$ | $0.906 \pm 0.053$ | $0.895 \pm 0.016$ | $0.862 \pm 0.254$ | $0.832 \pm 0.058$ |
| **NO-H2H** | | | | | | |
| HC vs. MLE | $0.901 \pm 0.439$ | $0.948 \pm 0.095$ | $0.970 \pm 0.019$ | $0.972 \pm 0.095$ | $0.933 \pm 0.154$ | $0.981 \pm 0.040$ |
| **Neutral decision support** | | | | | | |
| MLE | $\mathbf{0.753} \pm 0.056$ | $0.899 \pm 0.025$ | $0.896 \pm 0.044$ | $0.897 \pm 0.045$ | $0.901 \pm 0.025$ | $0.929 \pm 0.028$ |
| TML | $0.568 \pm 0.049$ | $0.775 \pm 0.084$ | $0.807 \pm 0.038$ | $0.868 \pm 0.012$ | $0.877 \pm 0.025$ | $\mathbf{1.000} \pm 0.000$ |
| HC | $0.740 \pm 0.540$ | $\mathbf{0.925} \pm 0.127$ | $\mathbf{0.933} \pm 0.064$ | $\mathbf{0.935} \pm 0.000$ | $\mathbf{0.945} \pm 0.349$ | $\mathbf{1.000} \pm 0.000$ |
| **Persuasive decision support** | | | | | | |
| MLE | $0.704 \pm 0.028$ | $0.900 \pm 0.017$ | $0.903 \pm 0.017$ | $0.903 \pm 0.017$ | $0.901 \pm 0.017$ | $0.919 \pm 0.016$ |
| TML | $0.906 \pm 0.011$ | $0.881 \pm 0.043$ | $0.863 \pm 0.044$ | $0.876 \pm 0.027$ | $0.877 \pm 0.076$ | $\mathbf{1.000} \pm 0.000$ |
| HC | $\mathbf{0.996} \pm 0.016$ | $\mathbf{0.995} \pm 0.000$ | $\mathbf{0.998} \pm 0.000$ | $\mathbf{0.996} \pm 0.016$ | $\mathbf{0.995} \pm 0.000$ | $0.995 \pm 0.032$ |

**Number of triplets.** We examine the effect of the number of triplets, showing the results in Fig. 6. We decrease number of triplets by powers of 2 and find that H2H preference towards human-compatible representations indeed declines as HC is less human-compatible with fewer training data. As for decision support, in neutral decision support HC performance declines and eventually approaches MLE representations except an outlier in the end, while in persuasive decision support HC performance is able to stay 100% even as the number of triplets declines.

**Additional details on weight generation.** We generate alignment scores by searching through weight combinations of the simulated human visual similarity metrics. We search the weights in powers of 2, from 0 to $2^{10}$, producing a sparse distribution of alignments (Fig. 7). Increasing search range to powers of 10 produces smoother distribution, but the weights are also more extreme and unrealistic. We note that the alignment distribution may vary across different datasets. In our experiments we choose weights and alignments to be as representative to the distribution as possible.

## C.3 ADDITIONAL DECISION BOUNDARIES

We create a variant of the VW dataset where the labels are populated by a linear separator. We refer to this dataset as VW-Linear (Fig. 8). We find the results are overall similar to the original VW data.

Table 6: Experiment results on VW. Models using 512-dimension embeddings; HC uses $\lambda = 0.8$ and filtered triplets.

| Alignments | 50% | 80% | 83% | 92% | 92.5% | 100% |
|---|---|---|---|---|---|---|
| Weights | [0,0,1,1] | [1,0,1,1] | [0,1,1,1] | [1,256,256,256] | [256,1,256,256] | [1,1,1,1] |
| **NI-H2H** | | | | | | |
| HC vs. MLE | $0.916 \pm 0.082$ | $0.869 \pm 0.217$ | $0.891 \pm 0.029$ | $0.879 \pm 0.066$ | $0.853 \pm 0.164$ | $0.828 \pm 0.138$ |
| **NO-H2H** | | | | | | |
| HC vs. MLE | $0.902 \pm 0.193$ | $0.944 \pm 0.093$ | $0.959 \pm 0.005$ | $0.956 \pm 0.090$ | $0.942 \pm 0.026$ | $0.969 \pm 0.034$ |
| **Neutral decision support** | | | | | | |
| MLE | $\mathbf{0.753} \pm 0.056$ | $\mathbf{0.899} \pm 0.025$ | $0.896 \pm 0.044$ | $0.897 \pm 0.045$ | $0.901 \pm 0.025$ | $0.929 \pm 0.028$ |
| TML | $0.568 \pm 0.049$ | $0.775 \pm 0.084$ | $0.807 \pm 0.038$ | $0.868 \pm 0.012$ | $0.877 \pm 0.025$ | $\mathbf{1.000} \pm 0.000$ |
| HC | $0.740 \pm 0.095$ | $0.894 \pm 0.111$ | $\mathbf{0.929} \pm 0.079$ | $\mathbf{0.960} \pm 0.032$ | $\mathbf{0.923} \pm 0.127$ | $\mathbf{1.000} \pm 0.000$ |
| **Persuasive decision support** | | | | | | |
| MLE | $0.704 \pm 0.028$ | $0.900 \pm 0.017$ | $0.903 \pm 0.017$ | $0.903 \pm 0.017$ | $0.901 \pm 0.017$ | $0.919 \pm 0.016$ |
| TML | $0.906 \pm 0.011$ | $0.881 \pm 0.043$ | $0.863 \pm 0.044$ | $0.876 \pm 0.027$ | $0.877 \pm 0.076$ | $\mathbf{1.000} \pm 0.000$ |
| HC | $\mathbf{0.998} \pm 0.032$ | $\mathbf{0.995} \pm 0.000$ | $\mathbf{0.998} \pm 0.000$ | $\mathbf{0.998} \pm 0.032$ | $\mathbf{0.995} \pm 0.000$ | $\mathbf{0.999} \pm 0.016$ |

Table 7: Experiment results on VW. Models use 512-dimension embeddings; HC uses $\lambda = 0.5$ and unfiltered triplets.

| Alignments | 50% | 80% | 83% | 92% | 92.5% | 100% |
|---|---|---|---|---|---|---|
| Weights | [0,0,1,1] | [1,0,1,1] | [0,1,1,1] | [1,256,256,256] | [256,1,256,256] | [1,1,1,1] |
| **NI-H2H** | | | | | | |
| HC vs. MLE | $0.921 \pm 0.015$ | $0.900 \pm 0.035$ | $0.920 \pm 0.023$ | $0.895 \pm 0.008$ | $0.867 \pm 0.034$ | $0.846 \pm 0.016$ |
| **NO-H2H** | | | | | | |
| HC vs. MLE | $0.951 \pm 0.034$ | $0.969 \pm 0.024$ | $0.991 \pm 0.002$ | $0.991 \pm 0.004$ | $0.958 \pm 0.010$ | $0.980 \pm 0.023$ |
| **Neutral decision support** | | | | | | |
| MLE | $\mathbf{0.753} \pm 0.056$ | $\mathbf{0.899} \pm 0.025$ | $\mathbf{0.896} \pm 0.044$ | $\mathbf{0.897} \pm 0.045$ | $\mathbf{0.901} \pm 0.025$ | $0.929 \pm 0.028$ |
| TML | $0.568 \pm 0.049$ | $0.775 \pm 0.084$ | $0.807 \pm 0.038$ | $0.868 \pm 0.012$ | $0.877 \pm 0.025$ | $\mathbf{1.000} \pm 0.000$ |
| HC | $0.603 \pm 0.051$ | $0.801 \pm 0.025$ | $0.848 \pm 0.053$ | $0.880 \pm 0.000$ | $0.880 \pm 0.081$ | $\mathbf{1.000} \pm 0.000$ |
| **Persuasive decision support** | | | | | | |
| MLE | $0.704 \pm 0.028$ | $0.900 \pm 0.017$ | $0.903 \pm 0.017$ | $0.903 \pm 0.017$ | $0.901 \pm 0.017$ | $0.919 \pm 0.016$ |
| TML | $0.906 \pm 0.011$ | $0.881 \pm 0.043$ | $0.863 \pm 0.044$ | $0.876 \pm 0.027$ | $0.877 \pm 0.076$ | $\mathbf{1.000} \pm 0.000$ |
| HC | $\mathbf{0.996} \pm 0.004$ | $\mathbf{0.999} \pm 0.004$ | $\mathbf{0.996} \pm 0.004$ | $\mathbf{0.996} \pm 0.004$ | $\mathbf{0.996} \pm 0.004$ | $0.997 \pm 0.004$ |

**Classification and triplet accuracy.** Table 9 shows classification and triplet accuracy of tuning $\lambda$, showing a similar trend to the previous experiment.

**H2H and decision support results** In Table 10 we present results with the best set of hyperparameter: filtered triplets, 512-dimension embedding, $\lambda = 0.5$. We show results for $\lambda = 0.2$ in Table 5 and $\lambda = 0.8$ in Table 6.

Similar to the experiment on VW square decision boundary, we see no clear relation between $\lambda$, embedding dimension and our evaluation metrics.

# D    HUMAN SUBJECT STUDY ON BUTTERFLIES V.S. MOTHS

## D.1    DATASET

Our BM dataset include four species of butterflies and moths including: Peacock Butterfly, Ringlet Butterfly, Caterpiller Moth, and Tiger Moth. An example of each species is shown in Fig 9.

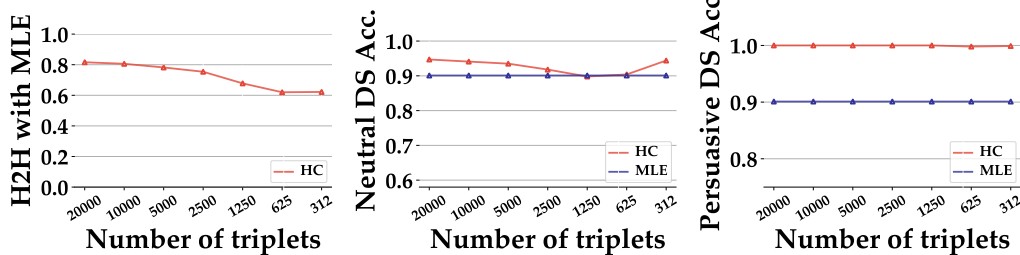

Figure 6: `HC` performance declines as the number of triplets decreases, but shows strong persuasive decision support accuracy even with very few triplets.

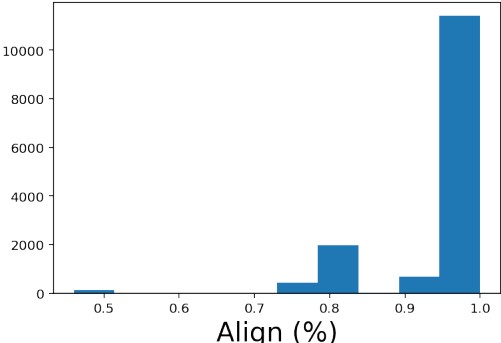

Figure 7: Histogram of alignments generated by searching informative weights in powers of 2.

## D.2   HYPERPARAMETERS

We use different controlling strength between classification and human judgment prediction, including $\lambda$s at 0.2, 0.5, and 0.8. We use the Adam optimizer (Kingma & Ba, 2014) with learning rate $1e - 4$. Our training batch size is 120 for triplet prediction, and 30 for classification. All models are trained for 50 epoches. The checkpoint with the lowest validation total loss in each run is selected for evaluations and applications.

## D.3   CLASSIFICATION AND TRIPLET ACCURACY

We present the test-time classification and triplet accuracy of our models in Table 13. Both `MLE` and `HC` achieve above 97.5% classification accuracy. `HC` in the 512-dimension unfiltered setting achieve 100.0% classification accuracy. Both `TML` and `HC` achieve above 70.7% triplet accuracy. Both `TML` and `HC` achieve the highest triplet accuracy in the 50-dimension unfiltered setting with triplet accuracy at 75.9% and 76.2% respectively. Filtering out class-inconsistent triplets removes 15.75% of the triplet annotations in this dataset.

We also evaluate the pretrained LPIPS metric (Zhang et al., 2018) on our triplet test set as baselines for learning perceptual similarity. Results with AlexNet backbone and VGG backbone are at 54.5% and 55.0% triplet accuracy respectively, suggesting that `TML` and `HC` provides much better triplet accuracy in this task.

## D.4   EFFECT OF TRIPLET AMOUNT AND TYPE

We evaluate the effect of the number of triplets on our models in Fig. 10. Similar to the VW experiments, H2H preference towards human-compatible representations and neutral decision support performance decrease as the number of triplets decreases. Human-compatible representations achieve strong persuasive decision support performance even with very few triplets.

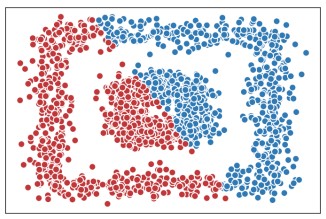

Figure 8: VW-Linear

Table 9: `HC` performance with different $\lambda$ on VW linear decision boundary data.

| Model | Classification accuracy | Triplet accuracy |
|---|---|---|
| `MLE` | $0.993 \pm 0.003$ | $0.673 \pm 0.014$ |
| `HC` $\lambda = 0.8$ | $0.988 \pm 0.032$ | $0.968 \pm 0.030$ |
| `HC` $\lambda = 0.5$ | $0.978 \pm 0.013$ | $0.966 \pm 0.007$ |
| `HC` $\lambda = 0.2$ | $0.978 \pm 0.032$ | $0.970 \pm 0.010$ |
| `TML` | N/A | $0.976 \pm 0.012$ |

Table 10: Experiment results on VW-Linear. Models use 512-dimension embeddings; `HC` uses $\lambda = 0.5$ and filtered triplets.

| Alignments | 56% | 84% | 95% | 98.5% |
|---|---|---|---|---|
| Weights | [0,1,1,1] | [1,0,1,1] | [1,1,1,1] | [32,256,1,1] |
| **NI-H2H** | | | | |
| `HC` vs. `MLE` | $0.913 \pm 0.023$ | $0.922 \pm 0.008$ | $0.899 \pm 0.020$ | $0.848 \pm 0.055$ |
| **NO-H2H** | | | | |
| `HC` vs. `MLE` | $0.932 \pm 0.034$ | $0.960 \pm 0.027$ | $0.921 \pm 0.013$ | $0.928 \pm 0.034$ |
| **Neutral decision support** | | | | |
| `MLE` | $0.778 \pm 0.084$ | $0.792 \pm 0.144$ | $0.839 \pm 0.130$ | $0.927 \pm 0.019$ |
| `TML` | $0.554 \pm 0.175$ | $0.770 \pm 0.318$ | $0.950 \pm 0.095$ | $0.914 \pm 0.075$ |
| `HC` | $\mathbf{0.841} \pm 0.053$ | $\mathbf{0.911} \pm 0.053$ | $\mathbf{0.967} \pm 0.009$ | $\mathbf{0.961} \pm 0.014$ |
| **Persuasive decision support** | | | | |
| `MLE` | $0.802 \pm 0.249$ | $0.815 \pm 0.151$ | $0.848 \pm 0.188$ | $0.953 \pm 0.051$ |
| `TML` | $0.473 \pm 1.016$ | $0.653 \pm 1.747$ | $0.441 \pm 0.016$ | $0.381 \pm 0.474$ |
| `HC` | $\mathbf{0.979} \pm 0.014$ | $\mathbf{0.977} \pm 0.009$ | $\mathbf{0.977} \pm 0.009$ | $\mathbf{0.978} \pm 0.013$ |

### D.5 MODEL EVALUATION WITH SYNTHETIC AGENT

We trained models with different configurations. We mainly discuss two factors: 1) filtering out class-inconsistent triplets or not; 2) a large dimension at 512 vs. a small dimension at 50 for the output representations. We also tried different hyperparameters such as different $\lambda$s that control the strength of the classification loss and triplet margin loss as well as different random seeds. We select the best `TML` / `HC` / `MLE` in each filtering-dimension configuration with the highest average of test classification accuracy and test triplet accuracies.

**Label accuracy and triplet accuracy.** As this task is relatively simple, both `MLE` and `HC` achieves test accuracy of above 97.5%. In fact, `HC` without filtering out class-inconsistent triplets achieved 100%. Note that `TML` cannot classify alone. As for triplet accuracy, as expected, both `HC` and `TML` outperform `MLE`. Dimensionality does not affect triplet accuracy, but filtering out class-inconsistent triplets decrease triplet accuracy (76.2% vs. 70.7% with 50 dimensions, 74.1% vs. 70.9% with 512 dimensions). This is because filtering creates a distribution shift of the triplet annotations, and limits the models' ability to learn general human visual similarity.

To run synthetic experiments for case-based decision support, we select the `TML` with the best test triplet accuracy as our synthetic agent, and then evaluate the examples produced by all representations. We do not show results of `TML` as we use it as the synthetic agent.

**Human-compatible representations is prefered over MLE representations in H2H.** We compare examples selected from different models in different configurations to examples selected by the `MLE` baseline with the same dimensionality.

Table 15 shows how often the synthetic agent prefers the tested model examples to baseline `MLE` examples. In all settings, the preference towards `HC` is above 50%, but not as high as those in our synthetic experiments with the VW dataset. Filtering out class-inconsistent triplets improves the preference for the nearest example with the predicted label, while hurting the preference for the nearest out-of-class example.

Table 11: Experiment results on VW-Linear. Models use 512-dimension embeddings; `HC` uses $\lambda = 0.2$ and filtered triplets.

| Alignments | 56% | 84% | 95% | 98.5% |
|---|---|---|---|---|
| Weights | [0,1,1,1] | [1,0,1,1] | [1,1,1,1] | [32,256,1,1] |
| **NI-H2H** | | | | |
| `HC` vs. `MLE` | $0.936 \pm 0.024$ | $0.921 \pm 0.008$ | $0.912 \pm 0.074$ | $0.856 \pm 0.034$ |
| **NO-H2H** | | | | |
| `HC` vs. `MLE` | $0.946 \pm 0.032$ | $0.974 \pm 0.032$ | $0.949 \pm 0.003$ | $0.934 \pm 0.029$ |
| **Neutral decision support** | | | | |
| `MLE` | $0.778 \pm 0.084$ | $0.792 \pm 0.144$ | $0.839 \pm 0.130$ | $0.927 \pm 0.019$ |
| `TML` | $0.554 \pm 0.175$ | $0.770 \pm 0.318$ | $0.950 \pm 0.095$ | $0.914 \pm 0.075$ |
| `HC` | $\mathbf{0.845} \pm 0.127$ | $\mathbf{0.880} \pm 0.127$ | $\mathbf{0.956} \pm 0.016$ | $\mathbf{0.956} \pm 0.111$ |
| **Persuasive decision support** | | | | |
| `MLE` | $0.802 \pm 0.249$ | $0.815 \pm 0.151$ | $0.848 \pm 0.188$ | $0.953 \pm 0.051$ |
| `TML` | $0.473 \pm 1.016$ | $0.653 \pm 1.747$ | $0.441 \pm 0.016$ | $0.381 \pm 0.474$ |
| `HC` | $\mathbf{0.974} \pm 0.016$ | $\mathbf{0.970} \pm 0.064$ | $\mathbf{0.968} \pm 0.064$ | $\mathbf{0.988} \pm 0.032$ |

Table 12: Experiment results on VW-Linear. Models use 512-dimension embeddings; `HC` uses $\lambda = 0.8$ and filtered triplets.

| Alignments | 56% | 84% | 95% | 98.5% |
|---|---|---|---|---|
| Weights | [0,1,1,1] | [1,0,1,1] | [1,1,1,1] | [32,256,1,1] |
| **NI-H2H** | | | | |
| `HC` vs. `MLE` | $0.906 \pm 0.122$ | $0.909 \pm 0.111$ | $0.882 \pm 0.135$ | $0.848 \pm 0.050$ |
| **NO-H2H** | | | | |
| `HC` vs. `MLE` | $0.926 \pm 0.021$ | $0.955 \pm 0.199$ | $0.936 \pm 0.053$ | $0.912 \pm 0.095$ |
| **Neutral decision support** | | | | |
| `MLE` | $0.778 \pm 0.084$ | $0.792 \pm 0.144$ | $0.839 \pm 0.130$ | $0.927 \pm 0.019$ |
| `TML` | $0.554 \pm 0.175$ | $0.770 \pm 0.318$ | $\mathbf{0.950} \pm 0.095$ | $0.914 \pm 0.075$ |
| `HC` | $\mathbf{0.824} \pm 0.175$ | $\mathbf{0.895} \pm 0.159$ | $\mathbf{0.950} \pm 0.032$ | $\mathbf{0.969} \pm 0.016$ |
| **Persuasive decision support** | | | | |
| `MLE` | $0.802 \pm 0.249$ | $0.815 \pm 0.151$ | $0.848 \pm 0.188$ | $0.953 \pm 0.051$ |
| `TML` | $0.473 \pm 1.016$ | $0.653 \pm 1.747$ | $0.441 \pm 0.016$ | $0.381 \pm 0.474$ |
| `HC` | $\mathbf{0.981} \pm 0.048$ | $\mathbf{0.964} \pm 0.206$ | $\mathbf{0.961} \pm 0.175$ | $\mathbf{0.978} \pm 0.064$ |

**Decision support simulations shows a large dimension benefits MLE representations but hurts unfiltered human-compatible representations in neutral decision support.** We also run simulated decision support with the `TML` synthetic agent. Table 16 shows decision support accuracy for different settings. `MLE` have both higher neutral decision support accuracy and persuasive decision support scores when we use a large dimension at 512. We hypothesize that for `MLE`, reducing dimension may force the network to discard dimensions useful for human judgments but keep dimensions useful for classification. We then use the 512-dimension `MLE` with the highest intrinsic evaluation scores as our `MLE` baseline in later studies.

For `HC`, neutral decision support accuracy are in general comparable to 87.5% score of the 512-dimension `MLE` baseline except unfiltered 512-dimension `HC` which has only 80%. We hypothesize that representations of large dimension may struggle more with contradicting signals between metric learning and supervised classification in the unfiltered settings. For persuasive decision support, `HC` achieves perfect scores in all settings.

Overall, to proceed with our human-subject experiments, we choose `HC` filtered with 50 dimensions as our best `HC` as it achieves a good balance between H2H and neutral decision support. For `MLE`, we choose the representation with 512 dimensions. We conduct head-to-head comparison between these

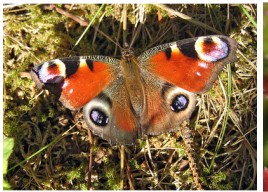 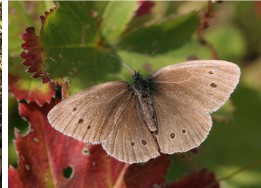 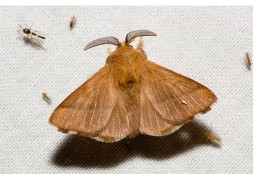 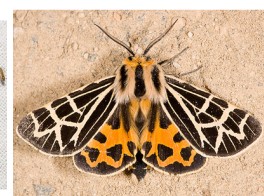

(a) Ringlet Butterfly (b) Peacock Butterfly (c) Caterpiller Moth (d) Tiger Moth

Figure 9: An example of each species in the BM dataset.

Table 13: Classification and triplet accuracy of BM models.

| Model | Classification accuracy | Triplet accuracy |
|---|---|---|
| **Dimension 50** | | |
| MLE | 0.975 | 0.610 |
| HC | 0.975 | 0.762 |
| HC-filtered | 0.975 | 0.707 |
| TML | N/A | 0.759 |
| TML-filtered | N/A | 0.721 |
| **Dimension 512** | | |
| MLE | 0.975 | 0.631 |
| HC | 1.000 | 0.741 |
| HC-filtered | 0.975 | 0.709 |
| TML | N/A | 0.748 |
| TML-filtered | N/A | 0.732 |

two representations. Our synthetic agent prefers `HC` in 70% of the nearest in-class examples and in 97.5% of the nearest out-of-class examples.

### D.6 INTERFACE

We present the screenshots of our interface at the end of the appendix. Our interface consists of four stages. Participants will see the consent page at the beginning, as shown in Fig 13. After consent page, participants will see task specific instructions, as shown in Fig 15. After entering the task, partipants will see the questions, as shown in Fig 16. We also include two attention check questions in all studies to check whether participants are paying attention to the questions. Following suggestions on Prolific, we design the attention check with explicit instructions, as shown in Fig 18. After finishing all questions, participants will reach the end page and return to Prolific, as shown in Fig 20. Our study is reviewed by the Institutional Review Board (IRB) at our institution (IRB22-0388).

### D.7 CROWDSOURCING

We recruit our participants on a crowdsourcing platform: Prolific (www.prolific.co) [April-May 2022]. We conduct three total studies: an annotation study, a decision support study, and a head-to-head comparison study. We use the default standard sampling on Prolific for participant recruitment. Eligible participants are limited to those reside in United States. Participants are not allowed to attempt the same study more than once.

**Triplet annotation study** We recruit 90 participants in total. We conduct a pilot study with 7 participants to test the interface, and recruit 83 participants for the actual collection of annotations. 3 participants fail the attention check questions and their responses are excluded in the results. We spend in total $76.01 with an average pay at $10.63 per hour. The median time taken to complete the study is 3'22".

**Decision support study** We recruit 161 participants in total. 3 participants fail the attention check questions and their responses are excluded in the results. We take the first 30 responses in each

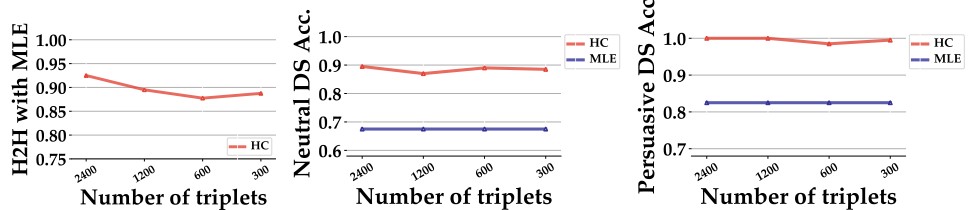

Figure 10: `HC` performance declines as the number of triplets decreases, but shows strong persuasive decision support accuracy even with very few triplets.

Table 15: BM H2H preference results with synthetic agent.

| Dimensions | 50 | 512 |
|---|---|---|
| **NI H2H with `MLE`** | | |
| `HC` | 0.838 | 0.575 |
| `HC` filtered | 0.863 | 0.725 |
| **NO H2H with `MLE`** | | |
| `HC` | 0.775 | 0.925 |
| `HC` filtered | 0.700 | 0.775 |

Table 16: BM decision support accuracy with synthetic agent.

| Dimensions | 50 | 512 |
|---|---|---|
| **Neutral Decision Support** | | |
| `MLE` | 0.675 | 0.875 |
| `HC` | 0.900 | 0.800 |
| `HC` filtered | 0.875 | 0.900 |
| **Persuasive Decision Support** | | |
| `MLE` | 0.825 | 0.875 |
| `HC` | 1.000 | 1.000 |
| `HC` filtered | 1.000 | 1.000 |

conditon to compile the results. We spend in total $126.40 with an average pay at $9.32 per hour. The median time taken to complete the study is 3'53".

**Head-to-head comparison study** We recruit 31 participants in total, where 1 participant fail the attention check questions and their responses are excluded in the results. We spend in total $24.00 with an average pay at $9.40 per hour. The median time taken to complete the study is 3'43".

# E    HUMAN SUBJECT STUDY ON CHEST X-RAYS

## E.1    DATASET

Our CXR dataset is constructed from a subset of the chest X-ray dataset used by Kermany et al. (2018), which had 5,232 images. We take a balanced subset of 3,166 images, 1,583 characterized as depicting pneumonia and 1,583 normal. The pneumonia class contains bacterial pneumonia and viral pneumoia images, but we do not differentiate them for this study. An example of each image class is shown in Fig 11.

## E.2    HYPERPARAMETERS

For CXR experiment, instead of ResNet-18 pretrained from ImageNet, we use a ResNet-18 finetuned on CXR classifcation as our CNN backbone, as we observe it provides better decision support simulation results. For training our `HC` model we use $\lambda$ of 0.5. We use the Adam optimizer (Kingma & Ba, 2014) with learning rate $1e-4$. Our training batch size is 16 for triplet prediction, and 30 for classification. All models are trained for 10 epoches. The checkpoint with the lowest validation total loss in each run is selected for evaluations and applications.

## E.3    CLASSIFICATION AND TRIPLET ACCURACY

We present the test-time classification and triplet accuracy of our models in Table 17. Both `MLE` and `HC` achieve above 95% classification accuracy. Both `TML` and `HC` achieve above above 65% triplet accuracy. Both `TML` model and `HC` achieve the highest triplet accuracy in the 512-dimension unfiltered setting with triplet accuracy at 69.1% and 72.2% respectively. Filtering out class-inconsistent triplets removes 20.69% of the triplet annotations in this dataset.

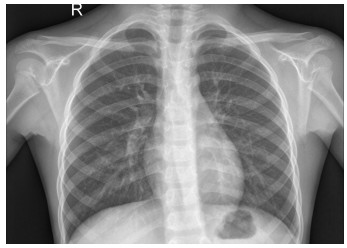
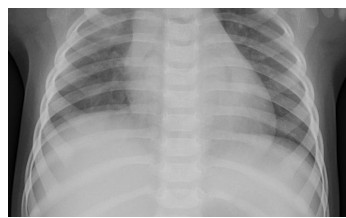
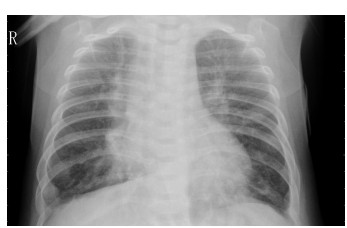

(a) Normal        (b) Bacterial pneumonia        (c) Viral pneumonia

Figure 11: An example of each image class in the CXR dataset.

Table 17: Classification and triplet accuracy of CXR models.

| Model | Classification accuracy | Triplet accuracy |
|---|---|---|
| **Dimension 50** | | |
| MLE | 0.973 | 0.571 |
| HC | 0.954 | 0.576 |
| HC-filtered | 0.955 | 0.574 |
| TML | N/A | 0.602 |
| TML-filtered | N/A | 0.587 |
| **Dimension 512** | | |
| MLE | 0.973 | 0.588 |
| HC | 0.968 | 0.602 |
| HC-filtered | 0.971 | 0.561 |
| TML | N/A | 0.618 |
| TML-filtered | N/A | 0.591 |

### E.4 MODEL EVALUATION WITH SYNTHETIC AGENT

Similar to the BM setting, we select the TML with the best test triplet accuracy as our synthetic agent, and then evaluate the examples produced by all representations. As table 18 shows, preference for HC over MLE in H2H is less significant compared to BM, likely due to the challenging nature of the CXR dataset. We still observe the patten that filtering improves H2H performance.

Table 19 shows decision support accuracy for different settings. All models benefit from a large dimension at 512. We observe consistent patterns such as filtering leading to better decision support.

### E.5 EFFECT OF TRIPLET AMOUNT AND TYPE

We evaluate the effect of the number of triplets on our models in Fig. 12. Similar to the BM experiments, H2H preference towards human-compatible representations and neutral decision support performance decrease as the number of triplets decreases. Human-compatible representations achieve strong persuasive decision support performance even with very few triplets.

### E.6 INTERFACE

Our CXR interface is mostly the same as our BM interface, except that we add basic chest X-ray instructions as participants may not be familiar with medical images. After the consent page at the beginning, participants will see basic chest X-ray instructions showing where the lungs and hearts. Then, they enter an multiple-choice attention check, as shown in Fig 14. The correct answer in "lungs and adjacent interfaces". Failing the attention check will disqualify the participant. After correctly answering the pre-task attention check, participants will see the same task specific instructions as in the BM studies, as shown in Fig 15. Screenshots of questions are shown in Fig 17. We also include two in-task attention check questions simlar to the BM study. Our study is reviewed by the Institutional Review Board (IRB) at our institution with study number that we will release upon acceptance to preserve anonymity.

Table 18: CXR H2H preference results with synthetic agent.

| Dimensions | 50 | 512 |
|---|---|---|
| **NI H2H with MLE** | | |
| HC | 0.536 | 0.675 |
| HC filtered | 0.472 | 0.599 |
| **NO H2H with MLE** | | |
| HC | 0.535 | 0.635 |
| HC filtered | 0.487 | 0.494 |

Table 19: CXR decision support accuracy with synthetic agent.

| Dimensions | 50 | 512 |
|---|---|---|
| **Neutral Decision Support** | | |
| MLE | 0.711 | 0.726 |
| HC | 0.742 | 0.779 |
| HC-filtered | 0.732 | 0.804 |
| **Persuasive Decision Support** | | |
| MLE | 0.881 | 0.882 |
| HC | 0.949 | 0.966 |
| HC-filtered | 0.948 | 0.946 |

Figure 12: HC performance declines as the number of triplets decreases, but shows strong persuasive decision support accuracy even with very few triplets.

### E.7 CROWDSOURCING

We recruit our participants on Prolific (www.prolific.co) [September 2022]. We conduct three total studies: an annotation study, a decision support study, and a head-to-head comparison study. We use the default standard sampling on Prolific for participant recruitment. Eligible participants are limited to those reside in United States. Participants are not allowed to attempt the same study more than once.

**Triplet annotation study** We recruit 123 participants in total. 20 partipants fail the pre-task attention check question and 3 participants fail the in-task attention check questions; their responses are excluded in the results. We spend in total $80.00 with an average pay at $10.70 per hour. The median time taken to complete the study is 3'22".

**Decision support study** We recruit 296 participants in total. 34 partipants fail the pre-task attention check question and 10 participants fail the in-task attention check questions; their responses are excluded in the results. We spend in total $221.67 with an average pay at $11.00 per hour. The median time taken to complete the study is 3'40".

**Head-to-head comparison study** We recruit 57 participants in total. 6 partipants fail the pre-task attention check question and 1 participants fail the in-task attention check questions; their responses are excluded in the results. We spend in total $40.00 with an average pay at $10.54 per hour. The median time taken to complete the study is 3'25".

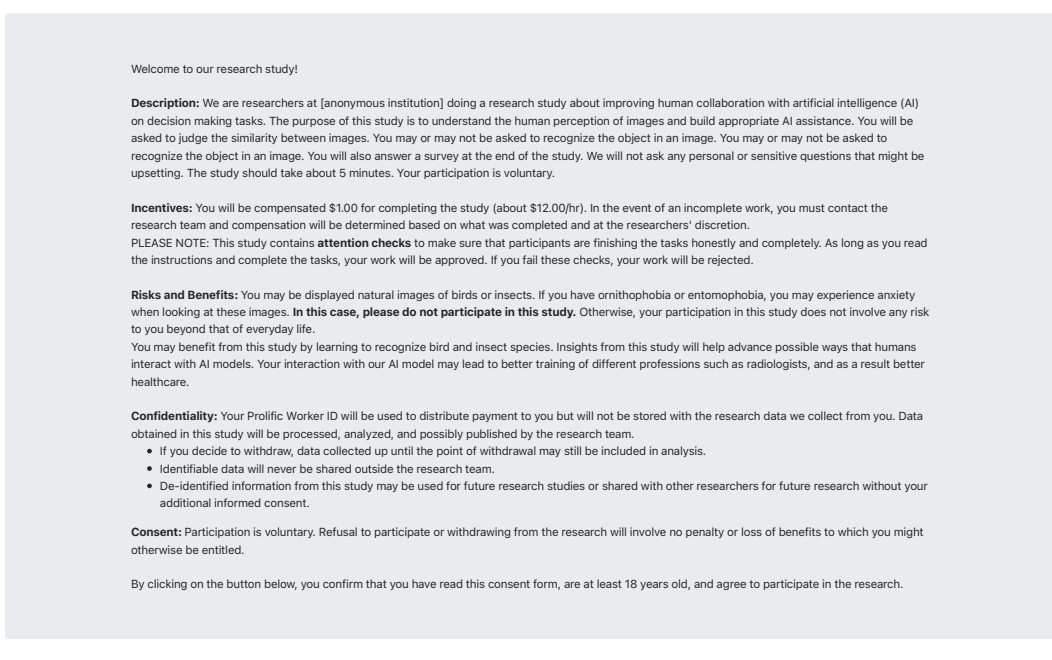

Figure 13: The consent form page on our interface.

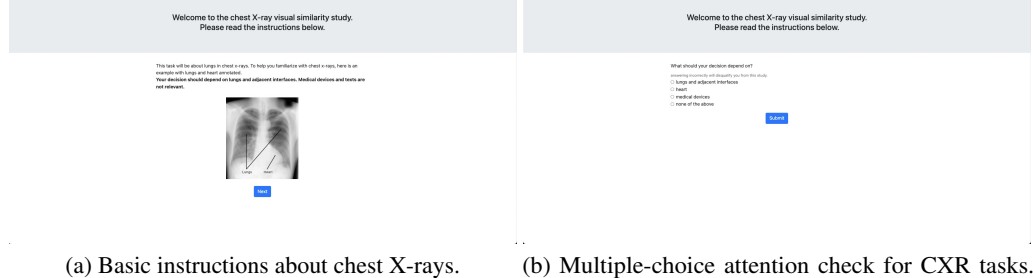

(a) Basic instructions about chest X-rays.

(b) Multiple-choice attention check for CXR tasks. The correct answer is "lungs and adjacent interfaces".

Figure 14: Pre-task instructions and attentions check for CXR tasks

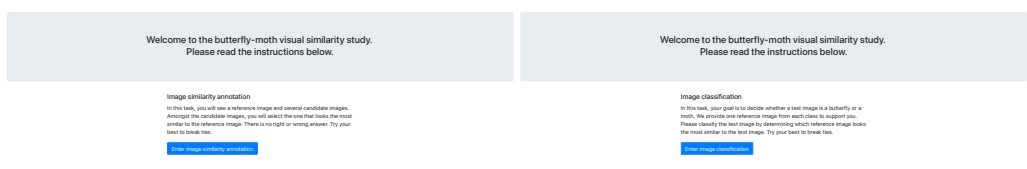

(a) The annotation and head-to-head comparision task instructions.

(b) The decision support task instructions.

Figure 15: The task-specific instruction page on our interface.

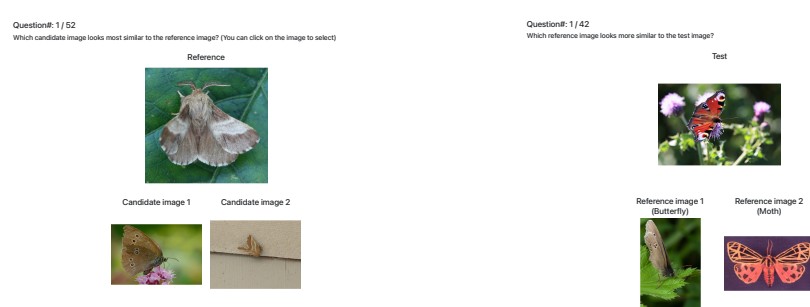

(a) The annotation and head-to-head comparision task questions.

(b) The decision support task questions.

Figure 16: The task-specific questions for BM.

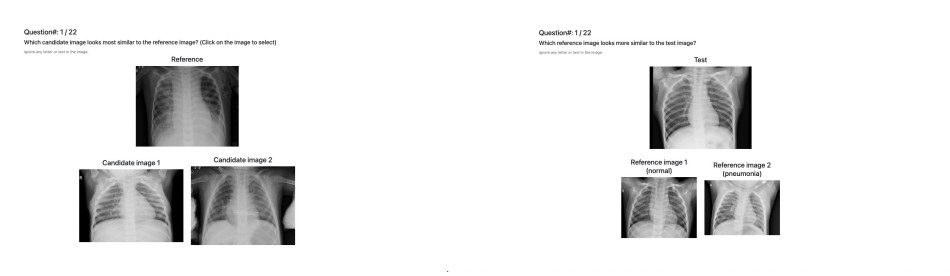

(a) The annotation and head-to-head comparision task questions.

(b) The decision support task questions.

Figure 17: The task-specific questions for CXR.

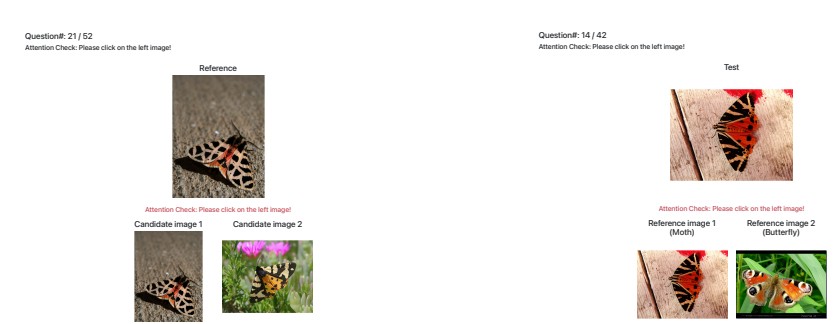

(a) The annotation and head-to-head comparision task attention check questions.

(b) The decision support task attention check questions.

Figure 18: The task-specific attention check questions for BM.

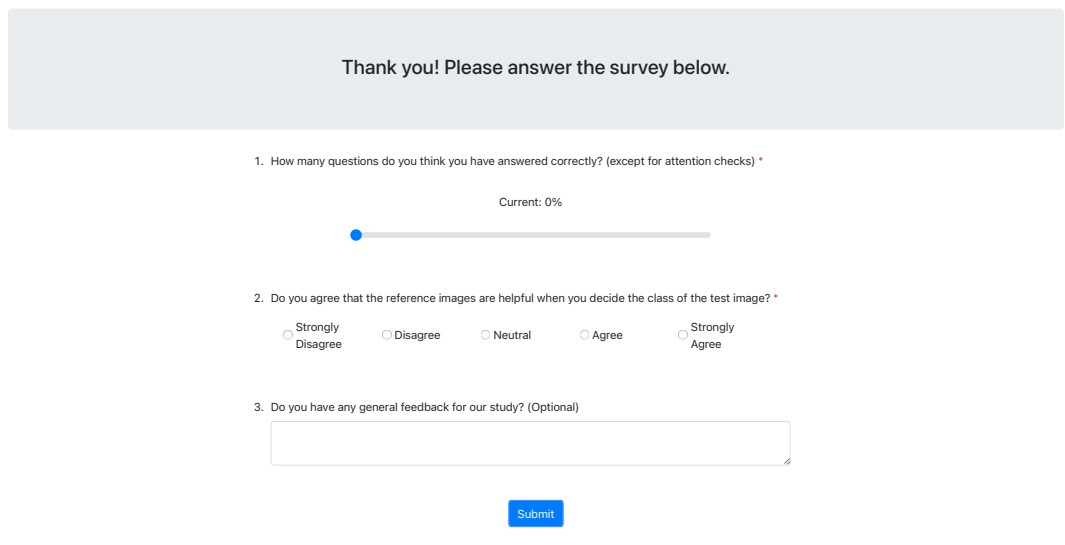

Figure 19: The survey page of the decision support task on our interface.

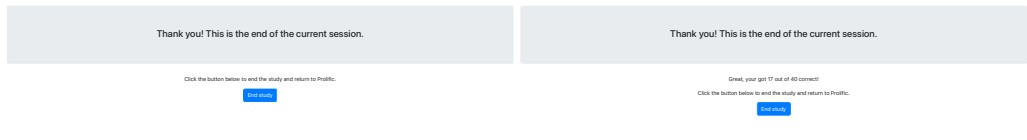

(a) The annotation and head-to-head comparision task end page.

(b) The decision support task end page.

Figure 20: The task-specific end page on our interface.

