# OpenReview forum: "Learning Human-Compatible Representations for Case-Based Decision Support"
_ICLR.cc/2023/Conference — ICLR 2023 poster_

### Official Review · Reviewer_J1hq · 2022-10-23

**Confidence:** 4
**Correctness:** 3
**Technical Novelty And Significance:** 2
**Empirical Novelty And Significance:** 3
**Recommendation:** 6

**Clarity, Quality, Novelty And Reproducibility:**

As mentioned above the paper is well written and easy to follow. The authors also tackle an important problem the gain in metric is significant attesting to the quality of the paper. As neither the MLE loss or the triplet loss is new, the proposed research is not very novel. The authors provide details regarding how the results can be reproduced.

**Strength And Weaknesses:**

Strengths:
The authors tackle an important problem. The proposed approach is simple yet intuitive. Combining two different losses is the correct approach to tackle this issue. Moreover, the experiments are well designed and consider various issues that might pop luck (like data augmentation gain). Finally, the paper is well written and easy to follow.

Weaknesses:
In cases where human judgement is necessary, often time the user would be an expert. Their judgment might vary a bit from the regular individuals. In such cases, the gain might be different from what is found in this paper. Furthermore, even though the paper is well written there are still some typos left. I would encourage the authors to fix those. The authors should also evaluate if the results hold for models other than ResNet. Furthermore, the authors can also consider how the approach can be used for other domains that is not image classification.

**Summary Of The Paper:**

In this paper the authors tackle the problem of case-based decision support. In the problem setting, a ML algorithm assists humans in making decisions on classification tasks by identifying an example from the training set that is similar to the unseen example. In the general case, the similarity between examples is determined by the metric space identified by the ML algorithm while training. However, this metric space can be very different from the space considered by the humans. This can cause the human judge to be misled. To tackle the issue, the authors propose a loss function which combines both MLE and a similarity loss. The similarity loss is derived from a triple learning problem (triplet margin loss). Results on both synthetic data and human experiments show that the proposed method performs much better than existing approaches.

**Summary Of The Review:**

The authors tackle an important problem and provide an important solution. Through multiple experiments they show that the proposed method is indeed better. However, the proposed approach is not significantly novel. Furthermore, more experiments are required with humans who are field experts and problems that are not in the image domain. These aspects reduces the impact of this paper a bit.

---

> ### Author Response · Authors · 2022-11-12
> **Responses to Reviewer J1hq**
>
> We appreciate the reviewer for the thoughtful comments! Below please find our detailed responses to the questions.
>
>
> ---
>
>
> #### **1) Gap between crowdworkers and experts**
> > In cases where human judgement is necessary, often time the user would be an expert. Their judgment might vary a bit from the regular individuals. In such cases, the gain might be different from what is found in this paper.
>
> **Response:**
> Applying our method to domain-experts is indeed our ultimate goal. We start with Prolific users here because recruiting experts is a challenging effort on its own. The positive results even with crowdworkers on challenging tasks such as chest X-rays are encouraging.
>
> Working with domain experts is indeed an important future direction. As the expertise level of the end users increases, HC should be able to learn a high-quality representation. Experts' improvement using our decision-support methods may be lower than lay peoples' due to experts' strong domain knowledge and baseline, but we would still expect our human-compatible representation to provide more effective decision support than MLE representations.
>
>
> #### **2) Typos**
> > Furthermore, even though the paper is well written there are still some typos left. I would encourage the authors to fix those.
>
> **Response:**
> Thank you for the suggestion! We have done multiple edit passes to fix typos. Please let us know if you spot anything else.
>
> #### **3) Additional models**
> > The authors should also evaluate if the results hold for models other than ResNet.
>
> **Response:** Model architecture was not the main concern of this paper. We chose ResNet due to its competitiveness and popularity. We leave the exploration of additional architecture to future work.
>
>
> #### **4) Application to other domains**
> > Furthermore, the authors can also consider how the approach can be used for other domains that is not image classification.
>
> **Response:**
> The high level idea of our approach is to learn a representation space compatible with human perceptions. We believe our work would generalize to other domains as long as human triplet annotations of input data are available for representation learning and case-based decision support could be applied. It would be an interesting problem to adapt our method to different tasks and data modalities while considering different forms of human perception signals and decision support formats.
>
>
> #### **5) Novelty**
> > the proposed approach is not significantly novel
>
> **Response:**
> While the proposed approach may not seem technically novel, our work makes two important contributions. First, we make a conceptual contribution by identifying the importance of learning human-compatible representations. We highlight that typical classification cannot learn good representations for decision support. Second, we develop novel evaluation setups (i.e., justification, neutral decision support, and persuasive decision support) to advance research on case-based decision support. Progress in evaluation is integral to any advances in techniques.

---

> > ### Comment · Reviewer_J1hq · 2022-11-14
> > **Response to author comments**
> >
> > I thank the authors for their responses. However, after reading the other reviews and the author responses currently I have decided not to update my scores. However, I will also keep an eye out for other reviewers opinions.

---

### Official Review · Reviewer_7Qs2 · 2022-10-24

**Confidence:** 3
**Correctness:** 3
**Technical Novelty And Significance:** 3
**Empirical Novelty And Significance:** 3
**Recommendation:** 6

**Clarity, Quality, Novelty And Reproducibility:**

Clarity: the work is easy to follow. Some elements are unclear in the introduction, but they are clarified on further reading.

Quality: the ideas are thoroughly examined and the experimental results are convincing. The human subject studies are appreciated and establish credibility for the proposed methods. However, the baseline is weak, and I wonder if there are other decision support methods that should be considered for comparison. See strengths and weaknesses section for other points.

Novelty: I am not sure. I am not familiar with decision support, and the authors do not provide much context for the work, such as prior methods that aim to address the same problem.

Reproducibility: I did not have time to read the appendix in detail, but the information required to reproduce the experiments seems to be present.

**Strength And Weaknesses:**

Strengths

The presentation is clear and the paper was easy and enjoyable to read. The work addresses an important and challenging problem and shows major improvement over baselines. The discussion of the ethics of decision support strategies and the difference between "neutral" and "persuasive" models is appreciated, and I get the impression that the work is well thought out and potential concerns are addressed in good faith. The results using human subjects are appreciated and make the claims credible.

Weaknesses (and questions and requests for clarification)

There is hardly any discussion of context and prior work in decision support. As a result, it is not clear to me how novel the methods are. Additionally, I'm not sure whether there is any previous work aimed at the particular decision support problems addressed in this work that would function as baselines for the proposed methods. If not, this should be explicitly stated, as the baselines used here are naive and fairly weak.

The purpose of the "justification" mode of decision support is not clear, and it should be better motivated. What is the goal? If the model's prediction is given, how would it improve human decision accuracy? Is it meant to earn trust? To be explainable? If the goal is related to trust or explainability, there seems to be an unstated assumption that training data with ground-truth labels can be blindly trusted and/or are themselves useful explanations, which seems insufficient. Whatever this goal is, is it reflected in the H2H score for evaluation?

I have concerns about the neutrality of filtering of class-inconsistent triplets. With respect to neutral decision support, you state that "the goal is not simply to maximize human decision accuracy, because one may use policies that intentionally show distant examples to nudge or manipulate human towards making a particular decision. Choosing nearest neighbors in each class is thus an attempt to present faithful and neutral evidence from the representation space so that humans can make their own decisions, hence preserving their agency. Therefore, the chosen nearest neighbors should be visually similar to the test instance by human perception [...]". The acknowledgment of this nuance is appreciated and shows that the ethical aspect of decision support is well considered. However, it seems to me that class-inconsistent triplet filtering is itself a roundabout way to allow the model to be "persuasive" rather than "neutral" by learning a representation that is only human-aligned when convenient, but not human-aligned when it does not result in a compelling argument for the model's predicted label. This could explain the gains in performance by using filtering for neutral decision support. I think that this should at least be addressed in the paper; ideally, the human subject experiments should also include results without filtering. I understand that this may not be possible in the timeline for revisions, however. Regardless, the neutral decision support results are impressive even with this concern.

The H2H score does not take into account the accuracy of the model, so it's hard to say that a HC model is useful for justification based on H2H alone. I would at least like to see the classification accuracy of the HC and MLE models. It would be even better if there is some way to consider accuracy as part of the the H2H score itself. For instance, one way might be to define the H2H score as, considering only data where the MLE prediction is correct, the fraction of such data that HC is preferable *and makes the correct prediction* so that cases that are well-justified but incorrect are not counted in its favor.

It's not clear how to interpret the relationship between task alignment and decision support results in the synthetic experiments.

It's not clear why we see a large disparity between H2H results for synthetic and human subject experiments. This should be addressed. Maybe it makes a difference whether the same human(s) (or synthetic humans) are used for both training data collection and testing?

Minor nit-picks:
- End of section 3, under "Head-to-head comparisons", the sentence "In addition to the typical justification for the predicted label, we also examine
that for the other class as those examples will be used in decision support": I cannot tell what the latter half of this statement is trying to say.
- Table 1, "Persuasive decision support", last column: the 1.000 in the middle row should be bold.


**Summary Of The Paper:**

The authors combine metric learning with classification to propose a data collection process and loss function for training models that are both accurate classifiers and learn representations that align with human perceptions of similarity in the data. The human-aligned representation can be used for case-based decision support, including justification, where a prediction is given and supported by similar training data with ground-truth labels, and neutral decision support, where no prediction is given, but the most similar labeled training data is given for each label to aid the human decision making process. Experiments on synthetic data with simulated humans show major improvement over a pure classifier for both justification and neutral decision support. Experiments on image classification with human subjects show minor or insignificant improvements in justification, but major and significant improvements in neutral decision support.

**Summary Of The Review:**

With the disclaimer that I don't have any background knowledge in decision support methods, I think this is a good paper with a few small but important concerns and missing pieces, detailed in the "Weaknesses" part of my "Strengths and Weaknesses" section. If these can be addressed, I will recommend it for acceptance.

---

> ### Author Response · Authors · 2022-11-12
> **Responses to Reviewer 7Qs2**
>
> We appreciate the reviewer for the thoughtful comments! Below please find our detailed responses to the questions.
>
> ---
>
> #### **1) Related work on decision support**
> > There is hardly any discussion of context and prior work in decision support. As a result, it is not clear to me how novel the methods are. Additionally, I'm not sure whether there is any previous work aimed at the particular decision support problems addressed in this work that would function as baselines for the proposed methods. If not, this should be explicitly stated, as the baselines used here are naive and fairly weak.
>
> **Response:**
> Prior work on AI decision-support and example-based explanations are ample, but work that used nearest-neighbor explanations are not directly comparable to our work (Wang and Yin, 2021; Lai and Tan, 2019; Nguyen et al., 2021). To the best of our knowledge, in the context of decision support, other work typically takes the representation as a given and the representation is often the MLE representation in our work. Therefore, we believe that our choice of baseline is appropriate.
>
> We have updated our related work section to situate our work in the literature of AI explanations and human-AI team decision making.
>
> #### **2) Purpose of justification is not clear**
> > The purpose of the "justification" mode of decision support is not clear, and it should be better motivated. What is the goal? If the model's prediction is given, how would it improve human decision accuracy? Is it meant to earn trust? To be explainable? If the goal is related to trust or explainability, there seems to be an unstated assumption that training data with ground-truth labels can be blindly trusted and/or are themselves useful explanations, which seems insufficient. Whatever this goal is, is it reflected in the H2H score for evaluation?
>
> **Response:**
> You are right in that the goal of justification is unclear and mixed. Justification is a minimal form of model explanation that can be used for many purposes; in our case we use it for AI-assisted decision making, but another common usage is explaining model decisions to non-decision-makers. The insufficiency of justification as decision support also motivates us to design additional evaluation methods like neutral and persuasive decision support.
>
> We consider a justification that is closer to the test instance to be a better explanation. This is reflected by the H2H score as it compares which of two justifications from different representations humans consider to be more similar.
>
>
> #### **3) Filtering**
> > I have concerns about the neutrality of filtering of class-inconsistent triplets. With respect to neutral decision support, you state that "the goal is not simply to maximize human decision accuracy…". The acknowledgment of this nuance is appreciated and shows that the ethical aspect of decision support is well considered. However, it seems to me that class-inconsistent triplet filtering is itself a roundabout way to allow the model to be "persuasive" rather than "neutral" by learning a representation that is only human-aligned when convenient, but not human-aligned when it does not result in a compelling argument for the model's predicted label. This could explain the gains in performance by using filtering for neutral decision support. I think that this should at least be addressed in the paper; ideally, the human subject experiments should also include results without filtering. I understand that this may not be possible in the timeline for revisions, however. Regardless, the neutral decision support results are impressive even with this concern.
>
> **Response:**
>
> The intent of filtering is to isolate the human intuition compatible with classification from the triplets to produce higher-quality representations. Filtering may lead the HC representation to be closer to MLE and exhibit more class separation, but this is aligned with our goal of creating a representation that is both human-aligned and does well in classification. Crucially, a representation being more class-consistent and aligned with the ground-truth is not equivalent to being more "persuasive" (e.g., neutral decision support still intends to show the best argument for each class). We view filtering as a design choice and our main point is not to argue for filtering over not filtering.
>
> Given our results on synthetic humans, we do not expect a substantial difference between filtering and not filtering in human evaluation. Since this is not our main goal, we chose one that performed slightly better in synthetic experiments. Nonetheless, you are right in that more research should be done on the effects of filtering.

---

> > ### Author Response · Authors · 2022-11-12
> > **Continued Responses to Reviewer 7Qs2**
> >
> > #### **4) H2H and classification accuracy**
> > > The H2H score does not take into account the accuracy of the model, so it's hard to say that a HC model is useful for justification based on H2H alone. I would at least like to see the classification accuracy of the HC and MLE models. It would be even better if there is some way to consider accuracy as part of the the H2H score itself. For instance, one way might be to define the H2H score as, considering only data where the MLE prediction is correct, the fraction of such data that HC is preferable and makes the correct prediction so that cases that are well-justified but incorrect are not counted in its favor.
> >
> > **Response:**
> > The classification accuracy for MLE is mentioned in the main paper, in the first paragraph of section 5: 97.5\% for BM and 97.3% for CXR. Classification accuracies for all models are shown in the appendix in Tables 3, 8, 12 and 15.
> >
> > Thank you for your suggestion on modifying H2H to consider accuracy; it would indeed provide a more precise measurement. However, given the high classification accuracies of our HC and MLE models, our H2H measurement should not be off by too much.
> >
> >
> > #### **5) Task alignment interpretation**
> > > It's not clear how to interpret the relationship between task alignment and decision support results in the synthetic experiments.
> >
> > **Response:**
> > Task alignment, as defined by 1-NN classification accuracy of the synthetic human's similarity metric, is set by tuning the informative features' weights. As stated in the main paper, decision support performance improves as the task alignment increases across all models, suggesting that decision support is easier when human similarity judgement is aligned with the classification task.
> >
> > The gap between using HC and MLE also increases as task alignment increases. When human similarity judgment is more aligned with the classification ground-truth, HC can learn a more consistent representation that is more beneficial to decision support.
> >
> >
> > #### **6) Disparaity in H2H results**
> > > it's not clear why we see a large disparity between H2H results for synthetic and human subject experiments. This should be addressed. Maybe it makes a difference whether the same human(s) (or synthetic humans) are used for both training data collection and testing?
> >
> > **Response:**
> > One reason may be that, since our H2H experiment consists of a test instance and two justifications, all of which are in the same class, humans may find the two justifications too similar and consider them as ties. Synthetic humans, however, can better discern small differences between the two justifications.
> >
> > #### **7) Clarity and format issues**
> > > End of section 3, under "Head-to-head comparisons", the sentence "In addition to the typical justification for the predicted label, we also examine that for the other class as those examples will be used in decision support": I cannot tell what the latter half of this statement is trying to say.
> > >
> > **Response:** We examine H2H for the examples that are in the same class as the predicted label (nearest in-class, NI) as well those in the opposing class (nearest out-of-class, NO). We revised this sentence to clarify this. We have also fixed the bolding issue in Table 1.
> >
> > #### **8) Additional decision support methods**
> > > However, the baseline is weak, and I wonder if there are other decision support methods that should be considered for comparison
> >
> > **Response:** Prior work has used nearest neighbors to choose examples for decision support, but, to the best of our knowledge, the question of which representation space to use has not been examined. The representation used in the classifier is commonly used and thus an appropriate baseline.
> >
> > One future direction would be to evaluate our human-compatible representation using other decision support methods like global examples, extracted features, etc.

---

> > > ### Author Response · Authors · 2022-12-05
> > > **Follow-up on Responses to Reviewer 7Qs2**
> > >
> > > Dear Reviewer 7Qs2,
> > >
> > > We wanted to follow up to see if our responses adequately addressed the concerns you raised in your review of our paper. We would be very grateful if you could provide any additional feedback or comments you may have. We are happy to provide further clarifications or explanations if needed.
> > >
> > > Thank you for your time and consideration!
> > >
> > > Paper1867 Authors

---

> > > > ### Comment · Reviewer_7Qs2 · 2022-12-05
> > > > **Reply from reviewer, and updated score**
> > > >
> > > > Authors,
> > > >
> > > > Thanks for your thorough reply, and I apologize for the lateness of my response.
> > > >
> > > > The reply at least partially addresses several of my concerns; I would ask to be sure that these are reflected in the text of the paper if they were not already, e.g. #3, #5, #6, and #8, for future readers who may have similar concerns. Assuming this is done, I'm going ahead and raising the overall score from 5 to 6.
> > > >
> > > > Thanks for your efforts to address my concerns and improve the paper in what ways you can given the limited time.

---

> > > > > ### Author Response · Authors · 2022-12-06
> > > > > **Reply to Reviewer 7Qs2**
> > > > >
> > > > > Thank you very much for the positive feedback! We agree these are valuable discussions that would benefit future readers. At the current stage we cannot update the paper submission, but we will make sure that we incorporate our responses in the text in the next version!

---

### Official Review · Reviewer_BaWn · 2022-10-25

**Confidence:** 3
**Correctness:** 4
**Technical Novelty And Significance:** 2
**Empirical Novelty And Significance:** 3
**Recommendation:** 6

**Clarity, Quality, Novelty And Reproducibility:**

The clarity of the contribution is good. The paper is fairly well-written. It could be improved if the introduction were explicit about the settings in which this method is applicable--for instance, that it requires human labelled data of triplet form. I also found the section on "machine teaching" in related work to be confusing. It would be helpful to clarify who is the teacher and the student.

The methodological novelty appears incremental as the method simply uses a linear combination of cross-entropy loss and a loss from prior work, the triple margin loss, for human compatibility. It would be helpful to clarify if there is more methodological novelty.

The paper offers novelty in its evaluation and empirical investigation. The paper provides a metric that assesses alignment with human assessment of similarity, the Head-2-Head (H2H) comparison. There's another evaluation metric under "neutral decision support" that is confusing since there are no details to explain what "accuracy of a synthetic human" refers to (perhaps this is in the appendix?). The H2H comparisons show that their proposed method better aligns with human assessment of similarity than the cross-entropy-loss minimizing model (the MLE). The plots contain error bars which is great.

The quality is acceptable but could be stronger if the paper provided more discussion and details about some key points:
- how much does the choice of metric impact the results. The paper states that Euclidean distance was chosen but were alternative metrics considered? It would be helpful to provide discussion of the benefits and limitations.
- how computationally intensive is the method?


**Strength And Weaknesses:**

+ The paper considers an important and impactful problem: how to build decision support tools that are interpretable in a faithful way to the prediction. The motivation was really strong and well-communicated.
+ The empirical evaluation is detailed and the results are interesting.

- The discussion on limitations is lacking. What situations may it be insufficient to provide only the most similar examples in each class? The ethics discussion at the end begins to discuss these limitations, but I think it would be useful to elaborate on this and put it some discussion in the main paper since it has implications for effectiveness. It could be illustrative for instance to give two examples: one setting that is well-suited to this method and another that is not.
- It would also be helpful to provide caveats about the empirical setup. For instance, what limitations result from having Prolific users (who are presumably not experts at reading x-rays) do the chest x-ray similarity assessments?

Please see next section for more on strengths and weaknesses with respect to novelty and clarity.

**Summary Of The Paper:**

This paper proposes a method for jointly learning a case-based predictive algorithm that maximizes accuracy and alignment with human assessments of similarity. The paper provides a loss function for this purpose and describes in detail experiments on synthetic and real-world data annotated by human labelers on Prolific. The real-world datasets describe chest x-rays (CXR) and moth vs. butterfly image classification.

**Summary Of The Review:**

This paper provides a simple solution to the well-posed problem: how to provide case-based decision support that aligns with human assessments of similarity while still being faithful to the predictions? The proposed method combines existing work to answer this, and the authors provide good empirical evaluations. The paper would be stronger if it provided more discussion on limitations and use cases.

---

> ### Author Response · Authors · 2022-11-12
> **Responses to Reviewer BaWn**
>
> We appreciate the reviewer for the thoughtful comments! Below please find our detailed responses to the questions.
>
> ---
>
> #### **1) Limitations of neutral decision support**
> > The discussion on limitations is lacking. What situations may it be insufficient to provide only the most similar examples in each class? The ethics discussion at the end begins to discuss these limitations, but I think it would be useful to elaborate on this and put it some discussion in the main paper since it has implications for effectiveness. It could be illustrative for instance to give two examples: one setting that is well-suited to this method and another that is not.
>
> **Response:**
> Thanks for the suggestion! We added a section on limitations to the appendix (section A), addressing the limitations of our decision-support methods. Unfortunately we cannot fit further ethics discussion in the main paper, but we did address some ethical concerns in section 2 when we first introduced neutral decision support.
>
> Providing the most similar examples in each class may not maximize decision accuracy. Thus, in cases where increasing human performance is pivotal, neutral decision support may be insufficient. On the other hand, neutral decision-support may be well-suited when the goal is to retain human agency and independent decision making.
>
> The tradeoff between human agency and performance is a common dilemma in human-AI teams. It is not our main concern to solve this dilemma, but we use neutral and persuasive decision-support to represent two ends of the spectrum, neutral retaining human agency and persuasive focusing on task performance (given that the AI outperforms human). We hope this invites more systematic studies in the future.
>
> Another instance where neutral decision support may not be well-suited is when the test instance is too close to data from other classes. Then the selected nearest-neighbors may appear too similar for humans to distinguish. In such cases, more sophisticated decision-support methods may be preferred.
>
> #### **2) Caveats about study with non-experts**
> > It would also be helpful to provide caveats about the empirical setup. For instance, what limitations result from having Prolific users (who are presumably not experts at reading x-rays) do the chest x-ray similarity assessments?
>
> **Response:**
> Our ultimate goal is to apply our method to domain-experts. We start with Prolific users and the positive results are encouraging, even for the challenging chest X-ray dataset. We hope these results could be used to convince and invite more domain experts to get involved and work towards an applicable system together in the future.
>
> Prolific users' performance indeed may not transfer to domain experts. A main limitation is that lay people have limited domain knowledge such as basic anatomy of body parts. This may lead to inconsistent decision making or inability to distinguish similar cases. We mitigate this limitation by providing an instruction and quiz section before our main study that provides basic information about how to examine chest X-rays.
>
>
> #### **3) Introduction not explicit about data requirements, machine teaching unrelated work confusing**
> > It could be improved if the introduction were explicit about the settings in which this method is applicable--for instance, that it requires human labelled data of triplet form. I also found the section on "machine teaching" in related work to be confusing. It would be helpful to clarify who is the teacher and the student.
>
> **Response:**
> We added a new sentence to make the use of triplet annotation more explicit. We also revised the machine teaching paragraph in the related work to situate our work in AI explanations and AI-assisted decision making.

---

> > ### Author Response · Authors · 2022-11-12
> > **Continued Responses to Reviewer BaWn**
> >
> > #### **4) Lack of methodological novelty**
> > > The methodological novelty appears incremental as the method simply uses a linear combination of cross-entropy loss and a loss from prior work, the triple margin loss, for human compatibility. It would be helpful to clarify if there is more methodological novelty.
> >
> > **Response:**
> > We would like to point out that our methodological contributions are twofold — not only a computational algorithm applied to a novel task, but also a novel evaluation framework. Our contribution in the algorithm highlights that typical classification cannot learn good representations for decision support. To evaluate representations used for decision support, we design two novel evaluation methods: H2H and decision support. H2H more directly compares two representations while decision support is closer to the actual decision making task.
> >
> > #### **5) Confusion in evaluation metric**
> > > There's another evaluation metric under "neutral decision support" that is confusing since there are no details to explain what "accuracy of a synthetic human" refers to (perhaps this is in the appendix?).
> >
> > **Response:**
> > "Accuracy of a (synthetic) human" refers to the synthetic human agent’s classification performance given our decision support methods (introduced in section 4). We construct a simulated human perceptual similarity metric so that the agent can make similarity judgments and thus predictions with our decision support just like a real human.
> >
> > To reduce confusion, we removed mentions of synthetic humans in section 3.
> >
> > #### **6) Clarity on how metric impact results: anything else besides euclidean? Computationally intensiveness?**
> > > The quality is acceptable but could be stronger if the paper provided more discussion and details about some key points:
> > -how much does the choice of metric impact the results. The paper states that Euclidean distance was chosen but were alternative metrics considered? It would be helpful to provide discussion of the benefits and limitations.
> > -how computationally intensive is the method?
> >
> > **Response:**
> > In our synthetic experiments, we tried using cosine distance but it resulted in models' worse performance. However, a future direction is to use similarity metrics in psychology literature (e.g., [Nosofsky, 1986](https://doi.org/10.1037//0096-3445.115.1.39)).
> >
> > Our method is not very computationally intense. The most notable computational work is in embedding triplet batches and the triplet loss computation. We do this by embedding the unique data points in each triplet batch, which is linear in the number of triplets and model size. The triplet loss computation is linear in the dimension, so the computation overhead is small.
> >
> > The additional cost is further mitigated by the sample efficiency of our approach. Our triplet learning is surprisingly data efficient. Synthetic experiments show our method works well with very few number of triplets and we added these results to the appendix (C.2, D.3, E.5).

---

> > > ### Author Response · Authors · 2022-12-05
> > > **Follow-up on Responses to Reviewer BaWn**
> > >
> > > Dear Reviewer BaWn,
> > >
> > > We wanted to follow up to see if our responses adequately addressed the concerns you raised in your review of our paper. We would be very grateful if you could provide any additional feedback or comments you may have. We are happy to provide further clarifications or explanations if needed.
> > >
> > > Thank you for your time and consideration!
> > >
> > > Paper1867 Authors

---

### Official Review · Reviewer_5Yxd · 2022-10-25

**Confidence:** 4
**Correctness:** 4
**Technical Novelty And Significance:** 2
**Empirical Novelty And Significance:** 3
**Recommendation:** 6

**Clarity, Quality, Novelty And Reproducibility:**

Quality: The paper is overall quite clear and easy to follow. However, it might help to provide an early visual example of what case based decision making appears like

Novelty: I think the paper has limited novelty, as effectively learning from triplet comparisons or human feedback is a well-studied problem, so the main contribution is showing the effectiveness of these representations for human-in-the-loop decision making. I think the author could discuss other explanation methods in the Related Works sections as well, as I think the paper's contribution falls more as another form of explanation and should thus be position with respect to those.

Reproducibility: Model training and crowdsourcing details are all provided.

**Strength And Weaknesses:**

S1: The paper tackles an interesting and well-defined problem of learning more human-compatible representations in the specific context of case-based decision support.

S2: The paper is very clearly written.

S3: The proposed approach, HC, strongly outperforms baselines in two real-world user study experiments.

W1: The details in 4.1 on the simulated human perceptual similarity metrics are very sparse and unclear, and it's not obvious whether this is a justifiable model of human perceptual judgements.

W2: The filtering of class-inconsistent triples is a bit concerning, as ideally the proposed method would be able to leverage information captured by such "noise" in annotations. It would be interesting to see how much filtering was performed for the experiments.

Q1: Could the authors clarify whether the model's predictions were shown to uses in the decision-support task and if not, why they were omitted, as case support is typically an augmented task (providing justification for a model's output)? It is not clear from the interface, and I'm not sure how relevant the results are if these outputs are omitted, as that would be a natural thing to include in a real world set-up.

**Summary Of The Paper:**

The paper focuses on the setting of case-based decision support where, in addition to receiving a model's prediction, a human decision-maker can also observe cases with similar-looking examples from the training set of the same label (justification), or provide similar looking examples with other labels (calibration). Specifically, the authors propose a loss function combining standard cross entropy loss with a triple margin loss that aligns representations closer together based on human judgements of similarity.

**Summary Of The Review:**

Overall, I think the paper addresses an interesting and relevant problem, and it's main weaknesses are novelty and clarity on certain experiment choices I mentioned in the review, and I'm happy to raise my score if the authors address them.

---

> ### Author Response · Authors · 2022-11-12
> **Responses to Reviewer 5Yxd**
>
> We appreciate the reviewer for the thoughtful comments! Below please find our detailed responses to the questions.
>
>
> ---
>
>
> #### **1) Synthetic human similarity metrics**
> > The details in 4.1 on the simulated human perceptual similarity metrics are very sparse and unclear, and it's not obvious whether this is a justifiable model of human perceptual judgements.
>
> **Response:**
> Thank you for pointing this out! We define our simulated human similarity metrics as a weighted euclidean distance based on the four visual features in the VW dataset. We added a formal definition of the simulated human metrics in section 4.1. More details on how we set the weights are in section C.2 in the appendix.
>
> We study the synthetic dataset for multiple reasons:
> 1. Collecting human triplets for larger datasets is expensive, so using a synthetic dataset allows us to experiment with a larger dataset. We experimented on VW with 2000 images and up to 40,000 triplets.
> 2. Synthetic experiments allows us to inspect the behavior of our algorithm in a controlled setting. We were able to explore a diverse set of experimental configurations, such as varying levels of disagreement between classification groundtruth and the human’s knowledge (task alignment score), in order to further investigate the strengths and limitations of our method.
>
> As the problem of modeling human perceptual judgments itself is a hard and ongoing problem in psychology and cognitive science, we do not attempt to come up with a perfect model in our simulated experiments. We are using our human model as a tool to simulate different scenarios we may encounter in practice and evaluate our proposed method.
>
> We acknowledge that our simulated human metrics may appear simple, but it serves as a reasonable basic model of human perceptual similarity. We have discussed among ourselves different ways to make this metric more realistic such as adding parameters to model human attention over different features, but we decided that the discussion might steer us towards another totally different work, and thus shift the focus of our work away from our original goal: *learning and evaluating human-compatible representations for decision support*. Results with the synthetic human are also complemented with studies with crowdworkers, where we observe similar behavior.
>
>
> #### **2) Filtering class-inconsistent triplets**
> > The filtering of class-inconsistent triples is a bit concerning, as ideally the proposed method would be able to leverage information captured by such "noise" in annotations. It would be interesting to see how much filtering was performed for the experiments.
>
> **Response:**
> This is a good question! The intent of filtering is to remove human perceptual signals that are not useful for this particular classification task in our collected triplets. These could include features that drive human similarity judgments but are not useful or even contradicting for the classification task, such as the background of an object in the image. We filter them out so that HC does not learn these different signals that may degrade the effectiveness of decision support.
>
> The portion of the triplets filtered out is 23.23% for VW, 15.75% for BM, and 20.69% for CXR. This information is available for VW in the last paragraph in 4.2. We added this information for BM and CXR to the sections D.3 and E.3 in the appendix.
>
> Note that we have indeed included results with *unfiltered triplets* in the Appendix. As shown in Figure 3 in the main paper and Table 10 in the appendix, unfiltered HC leads to slightly worse neutral decision support performance on the VW dataset.

---

> > ### Author Response · Authors · 2022-11-12
> > **Continued Responses to Reviewer 5Yxd**
> >
> > #### **3) Showing model predictions in decision support**
> > > Could the authors clarify whether the model's predictions were shown to uses in the decision-support task and if not, why they were omitted, as case support is typically an augmented task (providing justification for a model's output)? It is not clear from the interface, and I'm not sure how relevant the results are if these outputs are omitted, as that would be a natural thing to include in a real world set-up.
> >
> > **Response:**
> > Thank you for bringing up this question! The model predictions were not shown in decision support tasks. We added a sentence at the end of section 3 to clarify this.
> >
> > For both *neutral decision support* and *persuasive decision support*, we want to isolate the part of case-based decision making where the decision is completely determined by similarity between cases. Therefore, we intentionally omit the model predictions on the candidates for these two tasks. This allows us to focus on this single mechanism (i.e., decision making based on perceived visual similarity) and avoid influence from other possible cognitive processes. Showing labels may introduce additional mechanisms that could interact with and thus obfuscate the interpretation of results on our decision-support methods.
> >
> > For justification, showing predicted labels as suggested could make sense. That said, since both candidates come from the same class in our H2H evaluation, we believe that the effect of showing predicted labels in our interface is minimal.
> >
> >
> > #### **4) Visual example of case based decision making**
> > > However, it might help to provide an early visual example of what case based decision making appears like
> >
> > **Response:**
> > Thank you for the suggestion! Indeed, showing an early example can be very helpful. We could not fit examples of our interface in the main paper, so we include them in the appendix.
> >
> > #### **5) Limited novelty for triplet learning & related work on explanations**
> > > Novelty: I think the paper has limited novelty, as effectively learning from triplet comparisons or human feedback is a well-studied problem, so the main contribution is showing the effectiveness of these representations for human-in-the-loop decision making. I think the author could discuss other explanation methods in the Related Works sections as well, as I think the paper's contribution falls more as another form of explanation and should thus be position with respect to those.
> >
> > **Response:**
> > Although metric learning from human feedback and image classification are both well-studied problems, to the best of our knowledge, we believe our work is the first study to combine the two to bridge the gap between machine learning models and human perception, as well as the first work to systematically examine representations in case-based decision support with a thorough evaluation framework.
> >
> > Indeed, our work is closely related to human-AI team decision making and AI explanations. We have updated our related work section to situate our work in the literature of AI explanations.

---

> > > ### Author Response · Authors · 2022-12-05
> > > **Follow-up on Responses to Reviewer 5Yxd**
> > >
> > > Dear Reviewer 5Yxd,
> > >
> > > We wanted to follow up to see if our responses adequately addressed the concerns you raised in your review of our paper. We would be very grateful if you could provide any additional feedback or comments you may have. We are happy to provide further clarifications or explanations if needed.
> > >
> > > Thank you for your time and consideration!
> > >
> > > Paper1867 Authors

---

### Author Response · Authors · 2022-11-12
**Response to all reviewers**

We thank all the reviewers for their thoughtful reviews and suggestions. As the reviewers pointed out, our multi-task learning method effectively leads to human-compatible representations and shows great promises in providing case-based decision support. For example,

* The paper tackles an interesting and well-defined problem of learning more human-compatible representations in the specific context of case-based decision support (5Yxd).
* The proposed approach, HC, strongly outperforms baselines in two real-world user study experiments (5Yxd).
* The paper considers an important and impactful problem: how to build decision support tools that are interpretable in a faithful way to the prediction. The motivation was really strong and well-communicated (BaWn).
* The empirical evaluation is detailed and the results are interesting (BaWn).
* The presentation is clear and the paper was easy and enjoyable to read. The work addresses an important and challenging problem and shows major improvement over baselines (7Qs2).
* The authors tackle an important problem. The proposed approach is simple yet intuitive. Combining two different losses is the correct approach to tackle this issue. Moreover, the experiments are well designed and consider various issues that might pop luck (like data augmentation gain). Finally, the paper is well written and easy to follow (J1hq).


There were also some shared concerns raised, including the novelty of our approach (e.g., "The methodological novelty appears incremental" and "neither the MLE loss or the triplet loss is new"), insufficiency of our related work section ("the author could discuss other explanation methods in the Related Works sections"), and the choice of filtering class-inconsistent triplets ("concerns about the neutrality of filtering of class-inconsistent triplets"). We respond to these shared concerns below and summarize our changes made in the revision. We also provide more detailed responses to each reviewer.

---
### Novelty, related work and design choice

**Novelty**

As several reviewers mentioned, effective decision support represents an important challenge. Despite the extensive literature on explainable models, multi-task learning, and metric learning, to the best of our knowledge, our work is the first to leverage these existing computational tools to construct human-compatible representations for case-based decision support. We believe that this conceptual contribution is novel.

Our novelty also lies in our evaluation framework. To evaluate representations used for decision support, we design an evaluation framework with *H2H* comparison and *decision support* evaluation. H2H directly compares two representations while decision support is close to the actual decision making task.

Overall, we believe that our paper makes important contributions by developing a simple yet effective algorithm to learn human-compatible representations and designing an evaluation framework.

**Related work**

Case-based decision support is closely related to example-based explanations for AI predictions. We modified our related work section to discuss work on AI explanations and AI decision-support systems.

**Filtering class-inconsistent triplets**

This is a sensible question as filtering class-inconsistent triplets indeed removes some human perception information. The intent of filtering is to focus on relevant signals to the given classification task in order to provide better decision support. We did observe a small performance improvement with synthetic agents. Therefore, we use this in our human subject studies. It is useful to point out that we view filtering or not as a hyperparameter choice and the main point of our paper is not that filtering outperforms not filtering. We also include results for not filtering in the appendix. We encourage future work to further explore this direction.


---

### Summary of changes in the [revision](https://openreview.net/pdf?id=r0xte-t40I)

We clarified wordings with respect to filtering, H2H, and synthetic human similarity metrics. We also modified our related work section to include discussion about AI explanations and AI-assisted decision making. New texts are highlighted in blue.
* Section 1. Introduction: Clarified usage of triplet annotations.
* Section 3. Experimental Setup: Clarified filtering and justification.
* Section 4. Synthetic Experiment: Formalized simulated human similarity metric.
* Section 6. Related Work: Added AI explanations and AI-assisted decision making.
* Minor: Fixed number bold font in table 1 and other typos.
* Appendix A: Added a section on limitations.
* Appendix D and E: Added filtering details and number-of-triplets experiment plots.

These changes address most reviews in regards to content and clarity. We appreciate the valuable feedback.

---

### Decision · Program_Chairs · 2023-01-20

**Decision:**

Accept: poster

**Justification For Why Not Higher Score:**

Only marginal accpetances from the reviewers.

**Justification For Why Not Lower Score:**

Interesting work, deserves to be presented as poster at ICLR.

**Metareview: Summary, Strengths And Weaknesses:**

This paper has been assessed by four knowledgeable reviewers who independently, albeit very consistently, rated it as marginally acceptable. The paper is reasonably clearly written and it describes a strong contender method for human-in-the-loop learning from triplet comparisons. The reviewers point out that learning from triplets is not novel, but its application to interactive scenarios as presented can be considered novel. Some of the highlighted limitations, including lacking discussion on the limitations of the presented approach and suggestions for relevant use-cases, as well as some methodological concerns raised by the reviewers have been addressed by the authors during rebuttal process.

**Note From Pc:**

if the above contains the word "oral" or "spotlight" please see: "oral" presentation means -> notable-top-5% and "spotlight" means -> notable-top-25%. As stated in our emails, we are disassociating presentation type from AC recommendations